# Distributional Policy Evaluation: a Maximum Entropy approach to Representation Learning

**Riccardo Zamboni**
DEIB, Politecnico di Milano
Milan, Italy
riccardo.zamboni@polimi.it

**Alberto Maria Metelli**
DEIB, Politecnico di Milano
Milan, Italy
albertomaria.metelli@polimi.it

**Marcello Restelli**
DEIB, Politecnico di Milano
Milan, Italy
marcello.restelli@polimi.it

## Abstract

The Maximum Entropy (Max-Ent) framework has been effectively employed in a variety of Reinforcement Learning (RL) tasks. In this paper, we first propose a novel Max-Ent framework for policy evaluation in a distributional RL setting, named *Distributional Maximum Entropy Policy Evaluation* (D-Max-Ent PE). We derive a generalization-error bound that depends on the complexity of the representation employed, showing that this framework can explicitly take into account the features used to represent the state space while evaluating a policy. Then, we exploit these favorable properties to drive the representation learning of the state space in a Structural Risk Minimization fashion. We employ state-aggregation functions as feature functions and we specialize the D-Max-Ent approach into an algorithm, named *D-Max-Ent Progressive Factorization*, which constructs a progressively finer-grained representation of the state space by balancing the trade-off between preserving information (bias) and reducing the effective number of states, i.e., the complexity of the representation space (variance). Finally, we report the results of some illustrative numerical simulations, showing that the proposed algorithm matches the expected theoretical behavior and highlighting the relationship between aggregations and sample regimes.

## 1 Introduction

In Distributional Reinforcement Learning (D-RL) [Bellemare et al., 2023], an agent aims to estimate the entire distribution of the returns achievable by acting according to a specific policy. This is in contrast to and more complex than classic Reinforcement Learning (RL) [Szepesvári, 2010, Sutton and Barto, 2018], where the objective is to predict the expected return only. In recent years, several algorithms for D-RL have been proposed, both in evaluation and control settings. The push towards distributional approaches was particularly driven by additional flavors they can bring into the discourse, such as risk-averse considerations, robust control, and many regularization techniques [Chow et al., 2015, Brown et al., 2020, Keramati et al., 2020]. Most of them varied in how the distribution of the returns is modeled. The choice of the model was shown to have a cascading effect on how such a distribution can be learned, how efficiently and with what guarantees, and how it can be used for the control problem. Similarly to this tradition, this paper investigates the potential of looking into the entire distribution of returns to address the representation learning of the state-action spaces, that is to find a good feature representation of the decision-making space so as to make the

37th Conference on Neural Information Processing Systems (NeurIPS 2023).

overall learning problem easier, tenderly by reducing the dimensionality of such spaces. In particular, it points to answer the following research question:

Q1: *What tools can return distributions provide for distributional RL?*

In Section 3, we answer this methodological question, showing that it is possible to reformulate Policy Evaluation in a distributional setting so that its performance index is explicitly intertwined with the representation of the (state or action) spaces. More specifically, this work tackles Policy Evaluation in a distributional setting as a particular case of a distribution estimation problem and then applies the Maximum Entropy (Max-Ent) formulation of distribution estimation [Wainwright and Jordan, 2007]. In this way, it is possible to derive a novel framework for PE, which we name *Distributional Max-Ent Policy Evaluation* (D-Max-Ent PE), which inherits the many positives the Max-Ent framework offers. In particular, it allows the inclusion of constraints that the distribution needs to satisfy, usually called *structural constraints*, namely feature-based constraints acting over the support of the distribution. Such constraints then appear in the generalization-error bound for the Max-Ent problem, with a term related to the complexity of the family of features used. Unfortunately, traditional derivations of this bound introduce quantities that are bounded but unknown. We develop the analysis further to derive a more practical bound, containing quantities that are either estimated or known. In this way, the generalization-error bound shows a usual bias-variance trade-off that can be explicitly optimized by changing the feature functions adopted, while making the best use of the available samples. Thus, PE in a distributional setting is directly linked to representation learning.

Now, the RL literature proved that reducing the state space size while preserving the important features of the original state space is beneficial, namely with state-aggregation feature functions [Singh et al., 1994, Van Roy, 2006, Dong et al., 2020]. This is particularly true when high dimensionality can make learning slower and more unstable, as in classic RL in general, or when the learning process is almost unfeasible in small-samples regimes, as for D-RL, where learning the entire distribution of returns requires a large number of samples. Thus, motivated by these considerations, while D-Max-Ent Policy Evaluation allows for the use of any type of structural constraint, this work focuses on state-aggregation feature functions, and we exploit the first and more methodological result to answer a second more algorithmic question:

Q2: *How are representation learning and policy evaluation intertwined? Do distributional methods offer a new way to highlight and exploit this connection?*

To answer this question, in Section 4, we show that the generalization-error bound changes monotonically when the state-aggregation constraints are changed in a specific way, namely a finer-grained representation of the space. This is indeed what happens in the Structural Risk Minimization theory (SRM) [Vapnik, 1991]. Similarly, we develop a novel algorithm called *D-Max-Ent Progressive Factorization*, which exploits the proposed evaluation method to learn a representation of the state space soundly, i.e., trying to reduce a proxy of the generalization error bound in an SRM fashion, and allowing us to answer positively to the second research question as well. Finally, in Section 5 we verify through an illustrative numerical simulation whether the proposed algorithm matches the behaviors suggested by the theoretical analysis.

## 2 Preliminaries

### 2.1 Markov Decision Processes

A discrete–time finite Markov decision processes (MDP) [Puterman, 1994] is a tuple $\mathcal{M} :=
(\mathcal{S}, \mathcal{A}, P_{\mathcal{S}}, P_{\mathcal{R}}, \mu, \gamma)$, where $\mathcal{S}$ is a finite state space ($|\mathcal{S}| = S$), $\mathcal{A}$ is a finite action space ($|\mathcal{A}| = A$), $P_{\mathcal{S}} : \mathcal{S} \times \mathcal{A} \to \Delta(\mathcal{S})$ is the transition kernel, $P_{\mathcal{R}} : \mathcal{S} \times \mathcal{A} \to \Delta(\mathbb{R})$ is the reward distribution function, $\mu \in \Delta(\mathcal{S})$ is the initial-state distribution and $\gamma \in [0, 1)$ is discount factor.[1] A policy $\pi : \mathcal{S} \to \Delta(\mathcal{A})$ defines the behavior of an agent interacting with an environment, which goes as follows: starting from an initial state $S_0 \sim \mu$, an agent interacts with the environment through the policy $\pi$, generating a trajectory $\mathcal{H} = (S_t, A_t, R_t)_{t=0}^{\infty}$, which is a sequence of states, actions and rewards whose joint distribution is determined by the transition kernel, reward distribution, and the policy itself, i.e., $A_t \sim \pi(\cdot|S_t)$, $R_t \sim P_{\mathcal{R}}(\cdot|S_t, A_t)$, and $S_{t+1} \sim P_{\mathcal{S}}(\cdot|S_t, A_t)$.

---

[1]$\Delta(\mathcal{X})$ denotes the simplex of a space $\mathcal{X}$.

## 2.2 Value Functions and Distributions of Returns

Given an MDP $\mathcal{M}$ with discount factor $\gamma$, the *Discounted Return* is the sum of rewards received from the initial state onwards, discounted according to their time of occurrence:

$$\mathcal{G}^\pi(s) = \sum_{t=0}^\infty \gamma^t R_t | S_0 = s. \tag{1}$$

The *Value Function* of a given policy $\pi$ is the expectation of this quantity under the policy itself:

$$V^\pi(s) = \mathbb{E}[\mathcal{G}(s)] = \mathbb{E}\left[\sum_{t=0}^\infty \gamma^t R_t | S_0 = s\right]. \tag{2}$$

The *Return Distribution Function* $\eta^\pi$ of a given policy $\pi$ is a collection of distributions, one for each state $s \in \mathcal{S}$, where each element is the distribution of the random variable $\mathcal{G}^\pi(s)$:

$$\eta^\pi(s) = \mathcal{D}_{(s)}^\pi\left[\sum_{t=0}^\infty \gamma^t R_t | S_0 = s\right], \tag{3}$$

where $\mathcal{D}_{(s)}^\pi$ extracts the probability distribution of a random variable under the joint distribution of the trajectory.

The *Distributional Policy Evaluation Problem* then consists of estimating the return distribution function of Eq. (3) for a fixed policy $\pi$.

## 2.3 Maximum Entropy Estimation

*Maximum Entropy* (Max-Ent) methods [Dudík and Schapire, 2006, Wainwright and Jordan, 2007, Sutter et al., 2017] are density estimation methods that select the distribution that maximizes the uncertainty, i.e., the one with maximum entropy, where the entropy of a distribution $p \in \Delta(\mathcal{X})$ is defined as $H(p) := -\mathbb{E}_{X \sim p}[\log p(X)]$.[2] Additionally, they assume that the learner has access to a feature mapping $\mathcal{F}$ from $\mathcal{X}$ to $\mathbb{R}^M$. In the most general case, we may have $M = +\infty$. We will denote by $\Phi$ the class of real-valued functions containing the component feature functions $f_j \in \mathcal{F}$ with $j \in [M]$. A distribution $p$ is *consistent* with the true underlying distribution $p_0$ if

$$\mathbb{E}_{X \sim p}[f_j(X)] = \mu_j, \quad \forall j \in [M], \tag{4}$$

where

$$\mu_j := \mathbb{E}_{X \sim p_0}[f_j(X)] \tag{5}$$

In this case, we say that $p$ satisfies (in expectation) the structural constraints imposed by the features in $\mathcal{F}$. In practice, $p_0$ is not available and Max-Ent methods enforce empirical consistency over $N$ independent and i.i.d. observations $\mathcal{D} = \{x_1, \ldots, x_N\} \sim p_0$ with support in $\mathcal{X}$ by replacing the definition in Eq. (5) with

$$\hat{\mu}_j(\mathcal{D}) := \frac{1}{N} \sum_{i=1}^N f_j(x_i), \quad \forall j \in [M]. \tag{6}$$

The distribution $p$ is said to be consistent with the data $\mathcal{D}$ if it matches the empirical expectations. The empirical Max-Ent problem consists then of the following optimization problem

$$\begin{aligned} \max_{p \in \Delta(\mathcal{X})} \quad & H(p) \\ \text{s.t.} \quad & \mathbb{E}_{X \sim p}[f_j(X)] = \hat{\mu}_j, \quad \forall j \in [M], \end{aligned} \tag{7}$$

with the optimization problem in expectation differing just in the constraints (i.e., replacing constraint from Eq. (6) with the ones from Eq. (5)). It is well known that the optimal solution to the empirical

---

[2]With little abuse of notation, we will use the same symbol for the probability distribution and its p.d.f., which we assume to exist w.r.t. a reference measure.

Max-Ent problem in Eq. (7) is a distribution $p_\lambda \in \Delta(\mathcal{X})$ belonging to the class of exponential distributions parametrized by the parameters $\lambda$, namely:

$$p_\lambda(x) = \Phi_\lambda \exp \left( \sum_{j \in [M]} \lambda_j f_j(x) \right), \tag{8}$$

where $\Phi_\lambda := \int_{\mathcal{X}} \exp \left( \sum_{j \in [M]} \lambda_j f_j(x') \right) dx'$ is a normalization constant, which ensures that $p \in \Delta(\mathcal{X})$, and its log-transformation takes the name of log-partition function $A(\lambda) := \log \int_{\mathcal{X}} \exp(\sum_{j \in [M]} \lambda_j f_j(x)) dx$. The log-partition function defines the set of well-behaved distributions $\Omega = \{\lambda \in \mathbb{R}^M : A(\lambda) < +\infty\}$. At optimality, the parameters are defined as $\hat\lambda$ and correspond to the optimal Lagrangian multipliers of the dual of the empirical Max-Ent problem in Eq. (7). Now on, we will use $\hat p$ to identify $p_{\hat\lambda}$ for simplicity.

## 3  Distributional Policy Evaluation: A Max-Ent Approach

This section aims at answering the first research question Q1. The proposed approach turns distributional PE into a pure density estimation problem in a Max-Ent framework, called *Distributional Max-Ent Policy Evaluation*, as described in Algorithm 1. For this translation, the algorithm uses the distribution of returns $\eta$ as $p$, $N$-trajectory samples $\mathcal{H}_N = \{\mathcal{H}\}_{n=0}^N$ as data, and a fixed set of features functions $\mathcal{F}$ belonging to a function class $\Phi$. Note that to

---

**Algorithm 1** Distributional Max-Ent Policy Evaluation

---

**Require:** $(\mathcal{H}_N, \mathcal{F})$ $\quad \triangleright N$ trajectory samples, set of features functions
$\hat\eta = \underset{\eta}{\arg\max} \, H(\eta)$

s.t. $\mathbb{E}_{X \sim \eta}[f_j(X)] = \hat\mu_j(\mathcal{H}_N) \quad \forall j \in [M]$
$\quad \eta \in \Delta(\mathcal{X})$
**return** $\hat\eta$

---

do this, we need to slightly change the notation concerning the D-RL framework: $\eta$ will not be a $|\mathcal{S}|$-vector of distributions with support over $\mathbb{R}$, but rather a joint distribution over the whole support $\mathcal{X} = \mathcal{S} \times \mathbb{R}$. Turning PE into a Max-Ent problem has many upsides. First of all, the Max-Ent principle allows to deal with any kind of support $\mathcal{X}$, unifying continuous and discrete cases under the same framework; secondly, it does not require specifying a family of probability distributions to choose from; moreover, it implicitly manages the uncertainty by seeking a distribution as agnostic as possible, i.e., as close to the uniform distribution as possible. Finally, Max-Ent allows to include of structural constraints over the return distribution under many different flavors, both as in the standard value-function approximation methods [Van Roy, 2006] and as in more recent works based on statistical functionals acting over the return portion $\mathbb{R}$ of the support [Bellemare et al., 2023]. One of the possible limitations might be the requirement to have access to a batch of i.i.d. samples, but this is not necessarily restrictive: the result can be generalized for a single $\beta$-mixing sample path by exploiting blocking techniques [Yu, 1994, Nachum et al., 2019].

### 3.1  Generalization Error Bound

As previously said, the inner properties of Max-Ent allow for translating the results from density estimation methods to the distributional PE setting, and in particular, generalization-error bounds defined as KL-divergences.[3] Unfortunately, the generalization error bounds of traditional Max-Ent theory contain a conservative term that compares the solutions of the expectation and empirical Max-Ent problems, $\bar\eta, \hat\eta$ respectively, by taking the maximum between the 1-norm of the respective multipliers, namely $\max_{\lambda \in \{\bar\lambda, \hat\lambda\}} ||\lambda||_1$. This quantity is bounded yet unknown, making the result unpractical. In the following, we extend the previous results with a more practical bound containing $||\hat\lambda||_1$ instead of the maximum, requiring some additional assumptions about the expressiveness of the feature functions. This result is of independent interest and allows us to directly use the bound from an algorithmic perspective.

**Theorem 1** (Generalization Error Bound of D-Max-Ent PE). *Assume that the set of features $\mathcal{F}$ belong to the function class $\Phi$, which it is such that $\sup_{x \in \mathcal{X}, f \in \mathcal{F}} ||f(x)||_\infty = F < +\infty$ and that the*

---

[3]The KL-divergence between two distributions $p, q$ is defined as $KL(p||q) = \mathbb{E}_{x \sim p}[\log(p(x)/q(x))]$

*minimum singular value $\sigma_{\min}$ of the empirical covariance matrix of the features $\hat{\text{Cov}}(\mathcal{F})$ is strictly positive, namely $\sigma_{\min}(\hat{\text{Cov}}(\mathcal{F})) > 0$. Then, given a sample batch $\{x_1, \ldots, x_N\} \in \mathcal{X}^N$ of $N$ i.i.d. points drawn from the true distribution $\eta^\pi$, for any $\delta \in (0, 1)$, it holds with probability at least $1 - \delta$ that the solution to the sampled Max-Ent problem $\hat{\eta}$ satisfies the following:*

$$KL(\eta^\pi || \hat{\eta}) \precsim -H(\eta^\pi) + \tilde{\mathcal{L}}(\hat{\eta}) + B(\hat{\lambda}, \mathcal{F}, N, \delta) \tag{9}$$

$$\tilde{\mathcal{L}}(\hat{\eta}) = -\frac{1}{N} \sum_{i=0}^{N} \log \hat{\eta}(x_i) \tag{10}$$

$$B(\hat{\lambda}, \mathcal{F}, N, \delta) = 10||\hat{\lambda}||_1 \left( \mathcal{R}_N(\Phi) + F\sqrt{\frac{\log 1/\delta}{2N}} \right), \tag{11}$$

where $\precsim$ stands for the fact that the bound comprises additional terms that decrease at a higher rate in sample complexity and were therefore neglected. $H(\eta^\pi)$ and $\tilde{\mathcal{L}}(\hat{\eta})$, the empirical log-likelihood of the solution, form a bias term. The remaining term $B(\hat{\lambda}, \mathcal{F}, N, \delta)$ is a variance term depending on the multipliers characterizing the solution $\hat{\lambda}$, the number of samples, the confidence level $\delta$, and the feature class complexity as the empirical Rademacher complexity of the class $\mathcal{R}_N(\Phi)$ [Mohri et al., 2018].

## 3.2 Proof Sketch

Here we report the main steps of the proof of Th. 1. The interested reader can find the complete proof in Appendix A. First, define the set containing the solutions to the expected and sampled Max-Ent problems with $\mathcal{S} := \{\bar{\eta}, \hat{\eta}\}$, the related set for the multipliers $\Omega_\mathcal{S} := \{\bar{\lambda}, \hat{\lambda}\}$, and a quantity that will be central now on $h(x_1, \cdots, x_N) := \max_{\eta \in \mathcal{S}} |\mathbb{E}_{\eta^\pi}[\log \eta] - \frac{1}{N} \sum_i^N \log \eta(x_i)|$. Then, the building blocks of the error term $KL(\eta^\pi || \hat{\eta})$, namely $KL(\bar{\eta} || \hat{\eta})$ and $KL(\eta^\pi || \bar{\eta})$ are bounded by:

$$KL(\bar{\eta} || \hat{\eta}) \leq 2h(\cdot)$$
$$KL(\eta^\pi || \bar{\eta}) \leq -H(\eta^\pi) + \tilde{\mathcal{L}}(\hat{\eta}) + 3h(\cdot).$$

It is possible to show that:

$$h(\cdot) \leq 2 \sup_{\lambda \in \Omega_\mathcal{S}} ||\lambda||_1 \left( \mathcal{R}_N(\Phi) + F\sqrt{\frac{\log 1/\delta}{2N}} \right)$$

$$\sup_{\lambda \in \Omega_\mathcal{S}} ||\lambda||_1 \leq ||\hat{\lambda}||_1 + \sqrt{\frac{6M}{\sigma_{\min}(\hat{\text{Cov}}(\mathcal{F}))} h(\cdot)}.$$

The first inequality is obtained with standard methods as in van der Vaart and Wellner [1996], Dudley [1999], Koltchinskii and Panchenko [2002], Wang et al. [2013]. The second one is obtained by exploiting the intrinsic properties of the Max-Ent solution and by noting that it is possible to link $h(\cdot)$ with the Bregman divergence of the log-partition function $D_A(\bar{\lambda}, \hat{\lambda})$. One can see that the use of the second inequality introduces an additional assumption about the expressiveness of the feature functions, requiring the minimum singular value of the sampled covariance matrix $\sigma_{\min}(\hat{\text{Cov}}(\mathcal{F}))$ to be strictly positive. As a final step, setting $x = \sqrt{h(x_1, \cdots, x_N)}$ and combining the two previous inequalities yields a quadratic inequality:

$$\begin{cases} x^2 - bx - c \leq 0 \\ b = 2\sqrt{\frac{6M}{\sigma_{\min}(\hat{\text{Cov}}(\mathcal{F}))}} \left[ \mathcal{R}_N(\Phi) + F\sqrt{\frac{\log 1/\delta}{2N}} \right], \\ c = 2||\hat{\lambda}||_1 \left[ \mathcal{R}_N(\Phi) + F\sqrt{\frac{\log 1/\delta}{2N}} \right] \end{cases}$$

which is well-defined and solves for

$$h(x_1, \cdots, x_N) \precsim ||\hat{\lambda}||_1 \left( \mathcal{R}_N(\Phi) + F\sqrt{\frac{\log 1/\delta}{2N}} \right),$$

by neglecting higher-order terms. The statement of the theorem is then just a matter of combining all these results.

# 4 Distributional Representation Learning with State Aggregation

This section addresses the second research question Q2, namely how to use the bound in Th. 1 from an algorithmic perspective to automatically refine the features used to represent the state space in a principled way while performing D-Max-Ent PE. In particular, the focus is on a specific instance of feature functions for return distributions, namely state aggregation. More specifically, the state aggregation feature functions $\mathcal{F} = \{f_j\}_{j \in [M]}$ split the state space into $M$ disjoint subsets, one for each function, i.e., $\mathcal{S} = \cup_{j \in [M]} S_j$ and $S_j \cap S_{j'} = \emptyset$, $j, j' \in [M]$, $j \neq j'$, and gives back the associated return $g \in \mathbb{R}$, namely:

$$f_j : \mathcal{S} \times \mathbb{R} \to \mathbb{R} \\ f_j(s, g) = g \mathbb{1}_{[s \in S_j]}. \tag{12}$$

These features are bounded by the maximum return $G_{\max}$, while the empirical Rademacher complexity over $N$ samples of returns $\{(s_i, g_i)\}_{i \in [N]}$ can be directly computed as in Clayton [2014]:

$$\mathcal{R}_N(\Phi) = G_{\max} \sum_{j \in [M]} \sqrt{\hat{P}(S_j)}, \tag{13}$$

where $\hat{P}(S_j) = N_j/N$ and $N_j = |\{(g_i, s_i) : s_i \in S_j, i \in [N]\}|$. The decomposition of the Rademacher term into single terms leads to rewriting $B(\hat{\lambda}, \mathcal{F}, N, \delta)$ as in the following lemma.

**Lemma 1.** *For Distributional Max-Ent Evaluation with a state-aggregation feature class, the variance term $B(\hat{\lambda}, \mathcal{F}, N, \delta)$ is given by;*

$$B(\hat{\lambda}, \mathcal{F}, N, \delta) = 10||\hat{\lambda}||_1 G_{max} \left( \sum_{j \in [M]} \sqrt{\hat{P}(S_j)} + \sqrt{\frac{\log 1/\delta}{2N}} \right). \tag{14}$$

## 4.1 Representation Refinement: Progressive Factorization

State aggregation features are of interest due to the possibility of progressively refining the representation by increasing the factorization level, that is, by splitting a subset $S_j$ into further disjoint subsets. This refinement is called *progressive factorization* and is defined as follows.

**Definition 1** (Progressive Factorization). *For two sets of state aggregation feature functions, $\mathcal{F}, \mathcal{F}_j$, we say that $\mathcal{F}_j$ is a progressive factorization of $\mathcal{F}$, i.e., $\mathcal{F} \subset \mathcal{F}_j$, if $\mathcal{F} = \{f_1, \ldots, f_{j-1}, f_{j+1}, \ldots, f_M\} \cup \{f_j\}, \mathcal{F}_j = \{f_1, \ldots, f_{j-1}, f_{j+1}, \ldots, f_M\} \cup \{f_j^k\}_{k \in [K]}$ and the additional functions $\{f_j^k\}_{k \in [K]}$ are such that the corresponding subsets satisfy*

$$S_j = \bigcup_{k \in [K]} S_j^k, \quad S_j^k \cap S_j^{k'} = \emptyset, \; k, k' \in [K], \; k \neq k',$$

*where only non-degenerate class factorizations will be considered, meaning that the new subsets $S_j^k$ are non-empty.*

It is relevant for our interests that, in the case of progressive factorizations $\mathcal{F} \subset \mathcal{F}'$, the respective Max-Ent solutions enjoy the following monotonicity property

**Lemma 2** (Monotonicity). *The multipliers of the Max-Ent solutions $\hat{\lambda}, \hat{\lambda}'$ using $\mathcal{F} \subset \mathcal{F}'$ are such that*

$$||\hat{\lambda}||_1 \leq ||\hat{\lambda}'||_1. \tag{15}$$

This result is fully derived in Appendix C, and it ensures a monotonically increasing of all terms contained in the variance term of Eq. (11) since the complexity term is monotonically increasing by definition. On the other hand, the bias represented by Eq. (10) is guaranteed to decrease monotonically at finer levels of factorizations.

## 4.2 D-Max-Ent Progressive Factorization Algorithm

In summary, D-Max-Ent PE shows a generalization error bound whose quantities are either known or estimated and change monotonically between progressive factorizations. On these results, we build an algorithm called *D-Max-Ent Progressive Factorization*, shown in Algorithm 2, which iteratively constructs a sequence of feature sets $\mathcal{F}_0 \subset \mathcal{F}_1 \subset \ldots$ with progressive factorization while performing PE. The behavior of the algorithm is similar to what is done in Structural Risk-Minimization (SRM) [Vapnik, 1991], and it involves optimizing for a trade-off: the bias term (i.e., empirical risk) decreases by taking into account more complex features classes, while the variance term (i.e., the confidence interval) increases. The whole algorithm is then based on the progressive search for the new set of feature functions which reduces a proxy of the generalization error bound of D-Max-Ent PE:

$$\mathcal{J}(\hat{\eta}) = \beta \mathcal{L}(\hat{\eta}) + B(\hat{\lambda}, \mathcal{F}, N, \delta), \qquad (16)$$

and the procedure will continue until there are no further improvements in the trade-off. Due to the nature of the proxy function, the role of $\beta > 0$ is to regulate the tendency to factorize. Higher values of $\beta$ will increase the magnitude of the decreasing term, causing a boost in the tendency to factorize. On the other hand, lower values will further decrease the importance of this term, resulting in a lower tendency to factorization.

---

**Algorithm 2** Distributional Max-Ent Progressive Factorization

**Require:** $(\mathcal{H}_N, \mathcal{F}_0, \delta, \beta, K)$  $\triangleright$ $N$-trajectory samples, initial feature set, confidence level, boosting factor, factorization factor
1: Done $\leftarrow$ False, $i^* \leftarrow 0$
2: **while** not Done **do**
3:     $\mathcal{F} \leftarrow \mathcal{F}_{i^*}, M \leftarrow |\mathcal{F}|$
4:     $\hat{\eta} \leftarrow$ D-Max-Ent PE$(\mathcal{H}_N, \mathcal{F})$
5:     $\mathcal{J}(\hat{\eta}) \leftarrow \beta \mathcal{L}(\hat{\eta}) + B(\hat{\lambda}, \mathcal{F}, N, \delta)$
6:     $\{\mathcal{F}_j\}_{j \in [M]} \leftarrow$ Progressive Factor$(\mathcal{F}, K)$
7:     **for** $j \in [M]$ **do**
8:         $\hat{\eta}_j \leftarrow$ D-Max-Ent PE$(\mathcal{H}_N, \mathcal{F}_j)$
9:         $\mathcal{J}(\hat{\eta}_j) \leftarrow \beta \mathcal{L}(\hat{\eta}_j) + B(\hat{\lambda}_j, \mathcal{F}_j, N, \delta)$
10:        **if** $\mathcal{J}(\hat{\eta}_j) < \mathcal{J}(\hat{\eta})$ **then**
11:            $i^* \leftarrow j$
12:        **end if**
13:    **end for**
14:    **if** $\mathcal{F}_{i^*} == \mathcal{F}$ **then**
15:        Done $\leftarrow$ True
16:    **end if**
17: **end while**
18: **return** $\hat{\eta}_{i^*}$

---

Finally, the *Progressive Factor* function takes as input the list of feature functions and a factor $K$ and returns a list of progressively factored set of feature functions. More specifically, each element in $\{\mathcal{F}_j\}_{j \in [M]}$ corresponds to a progressive factorization of the feature $f_j$, factoring the related subset $S_j$ into $K$ disjoint subsets as in Definition 1. The new $K$ subsets $\{\mathcal{S}_k^j\}_{k \in [K]}$ are constructed in the worst-case scenario: the complexity term in Eq. (13) is maximized with partitions of a set leading to a uniform distribution of samples in each new partitioned subset, and since it is not possible to know in advance which samples will be contained in which new subset, one way is then to proceed with a uniform factorization. We decided to maintain the most agnostic approach over the set of possible features, but prior knowledge could be used to narrow down the partitions to consider.

## 5 Illustrative Numerical Simulations

This section reports the results of some illustrative numerical simulations that make use of Algorithm 2.

**Simulations Objectives** The objective of the simulations is to illustrate two essential features of the proposed method that were only suggested by the theoretical results. First of all, to analyze the outcome of performing policy evaluations with aggregated states at different sample regimes, by comparing the output of the proposed algorithm with some relevant baseline distributions. Secondly, the aim is to study the role of the boosting parameter $\beta$ and the sampling regime $N$, being the main hyper-parameters of Algorithm 2, in the tendency to factor the representation at utterly different sample regimes.

**MDP Instance Design** The effectiveness of the proposed approach is expected to be particularly evident in MDPs admitting a factored representation of the return distribution, namely the ones in which many states are nearly equivalent under the evaluation of a policy. This factorizability property is not uncommon in general since it is present in any environment with symmetries and Block-MDPs

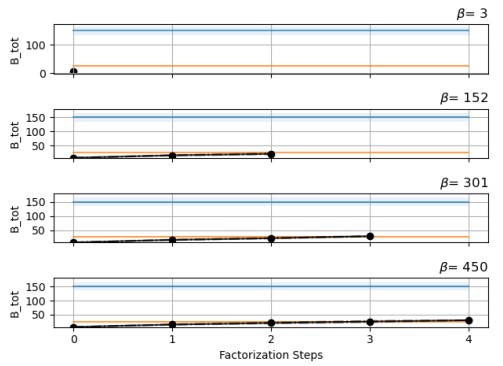

Figure 1: Bound Trend for different $\beta$ ($N = 50$)

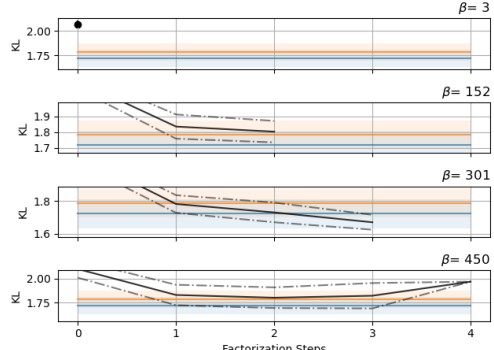

Figure 2: KL Trend for different $\beta$ ($N = 50$)

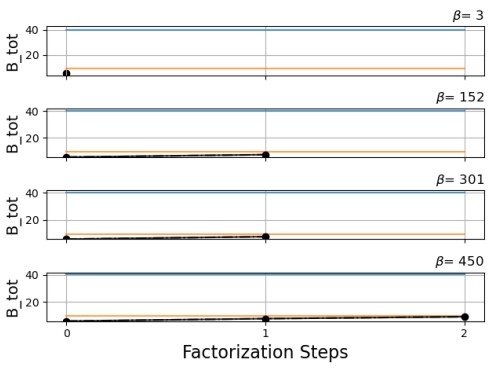

Figure 3: Bound Trend for different $\beta$ ($N = 1000$)

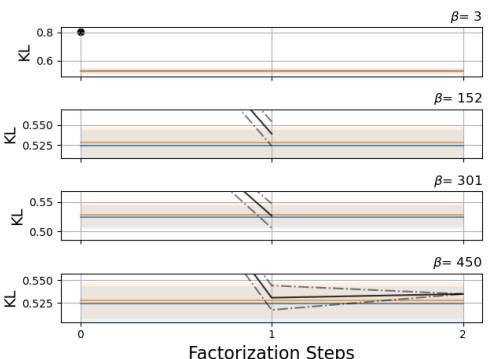

Figure 4: KL Trend for different $\beta$ ($N = 1000$)

[Du et al., 2019] as well. The MDP instance is then designed to be a Block-MDP indeed since it allows for better evaluate the simulation objectives: one would expect that operating on MDPs admitting a factored representation would allow for lower values of $\beta$ to be effective enough, while a higher level of boosting would force over-factorizations that are unnecessary, leading to no further improvement or even degradation of the results. The simulations are run on a rectangular GridWorld, with a height of $4$ and length of $8$, with traps on the whole second line and goals all over the top. A visualization of the setting can be seen in Appendix D. The policy is selected as a uniform distribution over the set of actions $\mathcal{A} = $ (up, left, right).

**Performance Indexes** The proposed MDP instance presents many upsides in terms of the interpretability of the output as well. First of all, it allows us to directly compute the true underlying return distribution with Monte-Carlo estimation. Secondly, it permits to compare of the output distribution of the algorithm with the result of performing plain Distributional Max-Ent Policy Evaluation (Algorithm 1) with two baseline representations: an **oracle factorization** that aggregates together states known to be equivalent under the policy, and in particular all the upper and lower states respectively; a **full factorization** that employs $|\mathcal{S}|$-singletons of states as representations, i.e., the most fine-grained representation possible. The comparison is made via two relevant quantities, the KL divergence with respect to the true distribution (the *bounded quantity*), and the total bound $\mathrm{B_{tot}} = \tilde{\mathcal{L}}(\hat{\eta}) + B(\hat{\lambda}, \mathcal{F}, N, \delta)$ (the *bounding quantity*). Finally, the value of the partition splitting $K$ is set to $2$, to reduce the exponential search space of all possible uniform partitions, the discount factor $\gamma$ is set to $0.98$ and the confidence $\delta$ to $0.1$, the results are averaged over $10$ rounds with the respective standard deviation.

**Results Discussion** The results of the simulations are reported from Fig. 1 to Fig. 4, with the quantity related to the oracle parametrization being in **orange**, while the ones related to the full parametrization being in **blue**. It is possible to notice that these two distributions have almost the same KL divergences with respect to the true return distribution (Fig. 2, 4), yet they highly differ

in the bound $B_{tot}$ (Fig. 1, 3) mostly due to the variance term, which is way higher in the case of full factorization. This suggests that the bound is indeed able to distinguish between the two. The plotting of the outputs of Algorithm 2 stops at the optimal number of factorization steps found for different values of $\beta$, namely at $\mathcal{F}_{i^\star}$. The plots should be read as follows: while the bound term $B_{tot}$ is expected to increase at each factorization step, the KL divergences with respect to the true return distribution should decrease as much as possible. In all cases, it is evident that the value of $\beta$ pushes towards a higher number of factorization steps, going from performing no factorization at all using low values ($\beta = 3$), to performing up to 4 factorization steps even in this simple scenario with higher values ($\beta = 450$), both at low and high sample regimes ($N \in \{50, 1000\}$). Furthermore, at higher sample regimes, it is possible to see how the higher quality of the estimation counteracts the action of $\beta$, and increasing it generally induces still fewer factorizations compared to the low sample regimes with same values of $\beta$, as in Fig. 3, 4. Finally, it is apparent that minimizing for Eq. (16) successfully decreases the KL divergence. Nonetheless, its values stop decreasing significantly after the first factorization, which splits the state space over the two rows and further factorizations might lead to performance degradation as well.

## 6  Discussion

In this section, we briefly discuss the literature related to this work and provide some concluding remarks about the results and future research paths.

### 6.1  Related Works

Our work relates to multiple fields. We now highlight the most relevant connections, while an exhaustive overview is beyond the scope of this paper.

**Distributional Reinforcement Learning**  D-RL has recently received much attention, both for the richness of flavors it admits [Rowland et al., 2019, 2021], and the surprising empirical effectiveness [Bellemare et al., 2020]. Our work tries to answer different research questions compared to traditional policy evaluation in D-RL and applies completely different techniques to derive the quantities of interest. Firstly, Bellman Operators and consequently contraction arguments cannot be applied since the estimation process is Monte-Carlo based. In this way, the sequential nature of the problem is not exploited, but we show that density estimation techniques do offer interesting properties nonetheless. Additionally, the employed indexes differ from traditional D-RL results. For example, the most recent bound for Q-TD [Rowland et al., 2023], shows a sub-linear term but the bound is made over the maximum Wasserstein distance, which cannot be directly related to the KL-divergence without further assumptions. Finally, distributional considerations have been employed in the field of function approximations as well. However, to the best of our knowledge, no other D-RL works explicitly address representation learning.

**Maximum Entropy and Feature Selection**  Max-Ent methods have a long and extensive literature in the density estimation field, which mostly focused on the general and algorithmic aspects of the method [Barron and Sheu, 1991, Dudík et al., 2004, Sutter et al., 2017]. Among the others, Cortes et al. [2015] proposed a Max-Ent regularized formulation for performing feature selection. Their method allocates different weights to different features to achieve an even better trade-off based on a combination. Due to this, their work differs from ours in the nature of the search space, which is not built progressively but is defined a priori. Additionally, their generalization bound is of the same nature as standard Max-Ent bounds and contains a $\sup_{\lambda \in \Omega} ||\lambda||_1$ term, which is bounded yet unknown. Finally, Mavridis et al. [2021] perform progressive state-aggregation through Max-Ent methods, but they try to optimize a different objective function based on state-dissimilarity.

### 6.2  Conclusions and Future Works

In our work, we presented in a D-RL framework a new policy evaluation approach based on Maximum Entropy density estimation, called *Distributional Max-Ent* Policy Evaluation, which benefits from the learning guarantees of Max-Ent and the generality of the setting, being able to enforce even complex feature families. We extended previous results and derived a practical formulation of the generalization error bound, which contains only estimated and known quantities of the problem. We then instantiated a particular class of features, namely state aggregation, and we proposed an algorithm

called *Distributional Max-Ent Progressive Factorization* to adaptively find a feature representation that optimizes for a proxy of the generalization error bound in a Structural Risk Minimization fashion. In this way, we showed that performing PE can indeed drive the learning of a reduced-dimension representation in the distributional setting. We then provided illustrative simulations showing the empirical behaviors of these approaches, while clarifying the links between some hyperparameters and the sample regime. Much of our analysis and theoretical guarantees straightforwardly extend to other feature classes, and an open question is to investigate other instances of features and settings that can benefit from the proposed framework. Future works will focus on the interaction between Temporal Difference (TD) distributional methods and representation learning in the proposed setting and on the existence of MDP instances that enjoy some relevant properties in the bias/variance trade-off along successive factorizations, leading to high performance or better error bounds.

## Acknowledgments

This paper is supported by PNRR-PE-AI FAIR project funded by the NextGeneration EU program.

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
