# A    Main Proof and Lemmas

In this section, we proceed to provide a proof of Theorem 1 of the main paper, together with some useful lemmas instrumental for proving it. Again, we define the set containing the solutions to the expected and sampled Max-Ent problems with $\mathcal{S} := \{\bar{\eta}, \hat{\eta}\}$, the related set for the multipliers $\Omega_{\mathcal{S}} := \{\bar{\lambda}, \hat{\lambda}\}$, which is a restriction of $\Omega = \{\lambda \in \mathbb{R}^M : A(\lambda) < +\infty\}$, and a quantity that will be central now on $h(x_1, \cdots, x_N) := \max_{\eta \in \mathcal{S}} |\mathbb{E}_{\eta^\pi}[\log \eta] - \frac{1}{N} \sum_i^N \log \eta(x_i)|$.

**Contribution Highlights**    The whole structure of the proof is built upon several intermediate results, of which some use standard techniques, and others are novel to this work. Here we report some comments to better clarify our contributions:

- Lemma 3 bounds the generalization-error with $h(\cdot)$, and it is based on the straighforward combination of Lemma 4 and Lemma 5.
- Lemma 4 introduces a slight modification to Wang et al. [2013] that is the use of the $\max_{\Omega_{\mathcal{S}}}$ over a finite set rather than $\sup_\Omega$ over the entire set of distributions. This will allow us to combine the result with the one of Lemma 5 and to deal with a simpler term, namely $h(x_1, \cdots, x_N)$ defined over the $\max$ instead of the $\sup$.
- Lemma 5 is a novel contribution, which was needed to obtain a practical form for the generalization error, compared to the intermediate result of Wang et al. [2013]. In this lemma as well $\max_{\Omega_{\mathcal{S}}}$ is employed, rather than $\sup_\Omega$.
- Lemma 6 uses standard techniques as can be found in van der Vaart and Wellner [1996], Dudley [1999], Koltchinskii and Panchenko [2002], but the analysis is again restricted to $\max_{\Omega_{\mathcal{S}}}$ thanks to the previous results.
- Lemma 7, Lemma 8 are novel results. They are needed to derive a practical generalization-error bound. Lemma 7 upper-bounds $||\bar{\lambda}||_1$ with $||\hat{\lambda}||_1$ by requiring additional constraints about the expressiveness of the feature functions. Lemma 8 uses this result to substitute $\max_{\lambda \in \Omega_{\mathcal{S}}} ||\lambda||_1$ with $||\hat{\lambda}||_1$.

As previously said, one of the main positives of this derivation is the ability to operate over $\max_{\eta \in \mathcal{S}}$ rather than $\sup_{\lambda \in \Omega}$. We will highlight the passages where this quantity is introduced with a ($\star$), and provide further comments.

## Initial step

First of all, we proceed in bounding the generalization error by bounding two sub-terms building it, that the following Lemma 3 will consist of a combination of two following lemmas, Lemma 4 and Lemma 5.

**Lemma 3.** *The generalization error between the true distribution and the Max-Ent solution of the sampled problem $\eta^\pi, \hat{\eta}$ (expressed as KL-divergence between the two distributions), given $N$ i.i.d. samples, can be bounded with the following quantity:*

$$KL(\eta^\pi||\hat{\eta}) \leq -H(\eta^\pi) + \tilde{\mathcal{L}}(\hat{\eta}) + 5 \max_{\eta \in \mathcal{S}} |\mathbb{E}_{\eta^\pi}[\log \eta] - \frac{1}{N} \sum_{j=0}^N \log \eta(x_j)|$$

*Proof.* As said, the result directly follows by considering that for the problem under consideration $KL(\eta^\pi||\hat{\eta}) = KL(\bar{\eta}||\hat{\eta}) + KL(\eta^\pi||\bar{\eta})$, since the two solutions correspond to the exact and sampled estimation problems. To bound the term on the right it is sufficient to bound the two terms on the left. We know that according to Lemma 4,

$$KL(\bar{\eta}||\hat{\eta}) \leq 2 \max_{\eta \in \mathcal{S}} |\mathbb{E}_{\eta^\pi}[\log \eta] - \frac{1}{N} \sum_{j=0}^N \log \eta(x_j)|$$

And according to Lemma 5

$$KL(\eta^\pi||\bar{\eta}) \leq -H(\eta^\pi) + \tilde{\mathcal{L}}(\hat{\eta}) + 3 \max_{\eta \in \mathcal{S}} |\mathbb{E}_{\eta^\pi}[\log \eta] - \frac{1}{N} \sum_{j=0}^N \log \eta(x_j)|$$

And the result directly follows. □

**Lemma 4.** *For the solutions of the exact and sampled Max-Ent problems, $\bar{\eta}$ and $\hat{\eta}$ respectively, it holds that*

$$KL(\bar{\eta}||\hat{\eta}) \leq 2 \max_{\eta \in \mathcal{S}} |\mathbb{E}_{\eta^\pi}[\log \eta] - \frac{1}{N} \sum_{j=0}^{N} \log \eta(x_j)|$$

*Proof.*

$$
\begin{aligned}
KL(\bar{\eta}||\hat{\eta}) &= KL(\eta^\pi||\hat{\eta}) - KL(\eta^\pi||\bar{\eta}) \\
&= (\mathbb{E}_{\eta^\pi}[\log \bar{\eta}] - \mathbb{E}_{\tilde{\eta}}[\log \bar{\eta}]) + (\mathbb{E}_{\tilde{\eta}}[\log \hat{\eta}] - \mathbb{E}_{\eta^\pi}[\log \hat{\eta}]) + (\mathbb{E}_{\tilde{\eta}}[\log \bar{\eta}] - \mathbb{E}_{\tilde{\eta}}[\log \hat{\eta}]) \\
&\leq 2 \max_{\eta \in \mathcal{S}=\{\bar{\eta},\hat{\eta}\}} |\mathbb{E}_{\eta^\pi}[\log \eta] - \frac{1}{N} \sum_{j=0}^{N} \log \eta(x_j)| + \frac{1}{N} \sum_{j=0}^{N} \log \frac{\bar{\eta}(x_j)}{\hat{\eta}(x_j)} \quad (\star) \\
&\leq 2 \max_{\eta \in \mathcal{S}} |\mathbb{E}_{\eta^\pi}[\log \eta] - \frac{1}{N} \sum_{j=0}^{N} \log \eta(x_j)|
\end{aligned}
$$

where the term $\frac{1}{N} \sum_{j=0}^{N} \log \frac{\bar{\eta}(x_j)}{\hat{\eta}(x_j)}$ is negative and then is removed from the bounding scheme.

$(\star)$ Here, Wang et al. [2013] bounded conservatively the first two terms $(\mathbb{E}_{\eta^\pi}[\log \bar{\eta}] - \mathbb{E}_{\tilde{\eta}}[\log \bar{\eta}]) + (\mathbb{E}_{\tilde{\eta}}[\log \hat{\eta}] - \mathbb{E}_{\eta^\pi}[\log \hat{\eta}])$ with the $\sup_{\lambda \in \Omega}$, yet we notice that the only two quantities of interest between which we are asked to maximize over are in the $\max_{\eta \in \mathcal{S}=\{\bar{\eta},\hat{\eta}\}}$. □

**Lemma 5.** *For the solutions of the Max-Ent problem in expectation $\bar{\eta}$ it is possible to bound the KL-divergence with respect to the true distribution $\eta^\pi$ with the following quantity*

$$KL(\eta^\pi||\bar{\eta}) \leq -H(\eta^\pi) + \tilde{\mathcal{L}}(\hat{\eta}) + 3 \max_{\eta \in \mathcal{S}} |\mathbb{E}_{\eta^\pi}[\log \eta] - \frac{1}{N} \sum_{j=0}^{N} \log \eta(x_j)|$$

*Proof.*

$$
\begin{aligned}
|\mathcal{L}_{\eta^\pi}(\bar{\eta}) - \tilde{\mathcal{L}}(\hat{\eta})| &= |\mathbb{E}_{\eta^\pi}[\log \bar{\eta}] - \frac{1}{N} \sum_{j=0}^{N} \log \hat{\eta}(x_j)| \\
&\leq |\mathbb{E}_{\eta^\pi}[\log \bar{\eta}] - \mathbb{E}_{\eta^\pi}[\log \hat{\eta}]| + |\mathbb{E}_{\eta^\pi}[\log \hat{\eta}] - \frac{1}{N} \sum_{j=0}^{N} \log \hat{\eta}(x_j)| \\
&\leq |KL(\eta^\pi||\bar{\eta}) - KL(\eta^\pi||\hat{\eta})| + \max_{\eta \in \mathcal{S}} |\mathbb{E}_{\eta^\pi}[\log \eta] - \frac{1}{N} \sum_{j=0}^{N} \log \eta(x_j)| \quad (\star) \\
&\leq 2 \max_{\eta \in \mathcal{S}} |\mathbb{E}_{\eta^\pi}[\log \eta] - \frac{1}{N} \sum_{j=0}^{N} \log \eta(x_j)| + \max_{\eta \in \mathcal{S}} |\mathbb{E}_{\eta^\pi}[\log \eta] - \frac{1}{N} \sum_{j=0}^{N} \log \eta(x_j)| \\
&\leq 3 \max_{\eta \in \mathcal{S}} |\mathbb{E}_{\eta^\pi}[\log \eta] - \frac{1}{N} \sum_{j=0}^{N} \log \eta(x_j)|
\end{aligned}
$$

$(\star)$ Again, due to the conservative bound in Lemma 4, Wang et al. [2013] maintained the same quantity in this bound for later simplifications. We apply a tighter bound of $\max_{\eta \in \mathcal{S}} |\mathbb{E}_{\eta^\pi}[\log \eta] - \frac{1}{N} \sum_{j=0}^{N} \log \eta(x_j)|$ to $|\mathbb{E}_{\eta^\pi}[\log \hat{\eta}] - \frac{1}{N} \sum_{j=0}^{N} \log \hat{\eta}(x_j)|$.

It follows that it is possible to write

$$|\mathcal{L}_{\eta^\pi}(\bar\eta) - \tilde{\mathcal{L}}(\hat\eta)| = |KL(\eta^\pi || \bar\eta) + H(\eta^\pi) - \tilde{\mathcal{L}}(\hat\eta)|$$

$$|KL(\eta^\pi || \bar\eta) - (-H(\eta^\pi) + \tilde{\mathcal{L}}(\hat\eta))| \leq 3 \max_{\eta \in \mathcal{S}} |\mathbb{E}_{\eta^\pi}[\log\eta] - \frac{1}{N}\sum_{j=0}^{N}\log\eta(x_j)|$$

$$||KL(\eta^\pi || \bar\eta)| - |(-H(\eta^\pi) + \tilde{\mathcal{L}}(\hat\eta))|| \leq 3 \max_{\eta \in \mathcal{S}} |\mathbb{E}_{\eta^\pi}[\log\eta] - \frac{1}{N}\sum_{j=0}^{N}\log\eta(x_j)|$$

which proves the result. □

**Intermediate Step**

As suggested by the previous considerations, everything boils down to being able to bound the term $h(x_1, \cdots, x_N) := \max_{\eta\in\mathcal{S}} |\mathbb{E}_{\eta^\pi}[\log\eta] - \frac{1}{N}\sum_{j=0}^{N}\log\eta(x_j)|$. To do this, we used standard techniques to derive the following intermediate step, where we can bound the quantity of interest which depends on the supremum between distributions $\max_{\eta\in\mathcal{S}} |\cdot|$ with a quantity depending on the supremum between their respective parameters $\lambda \in \Omega_{\mathcal{S}}$, namely $\sup_{\lambda\in\Omega_{\mathcal{S}}} ||\lambda||_1$.

**Lemma 6.** *The supremum difference between the expected log-likelihood and the sampled one, taken over the expected and sampled solutions in $\mathcal{S} = \{\bar\lambda, \hat\lambda\}$, is defined as $h(x_1, \cdots, x_N) := \max_{\eta\in\mathcal{S}} |\mathbb{E}_{\eta^\pi}[\log\eta] - \frac{1}{N}\sum_{j=0}^{N}\log\eta(x_j)|$ and it can be bounded by*

$$\max_{\eta\in\mathcal{S}} |\mathbb{E}_{\eta^\pi}[\log\eta] - \frac{1}{N}\sum_{j=0}^{N}\log\eta(x_j)| \leq 2\sup_{\lambda\in\Omega_{\mathcal{S}}} ||\lambda||_1 \mathcal{R}_N(\Phi) + 2\sup_{\lambda\in\Omega_{\mathcal{S}}} ||\lambda||_1 F\sqrt{\frac{\log 1/\delta}{2N}}$$

*with $F = \sup_{f\in\mathcal{F}} ||f||_\infty$.*

*Proof.* We define

$$h(x_1, \ldots, x_N) = \max_{\eta\in\mathcal{S}} |\mathbb{E}_{\eta^\pi}[\log\eta] - \frac{1}{N}\sum_{j=0}^{N}\log\eta(x_j)|$$

$$= \sup_{\lambda\in\Omega_{\mathcal{S}}, f\in\mathcal{F}} |\mathbb{E}_{\eta^\pi}\langle\lambda, f(x)\rangle - \frac{1}{N}\sum_{j=0}^{N}\langle\lambda, f(x)\rangle|$$

Then by exploiting the definition of the function, we study the differences induced by changing one sample from $x_k$ to $x_k'$

$$|h(x_1, \ldots, x_M) - h(x_1, \ldots, x_k', \ldots, x_M)| =$$

$$= |\sup_{\lambda\in\Omega_{\mathcal{S}}, f\in\mathcal{F}} |\mathbb{E}_{\eta^\pi}\langle\lambda, f(x)\rangle - \frac{1}{N}\sum_{j=0}^{N}\langle\lambda, f(x)\rangle|$$

$$- \sup_{\lambda\in\Omega_{\mathcal{S}}, f\in\mathcal{F}} |\mathbb{E}_{\eta^\pi}\langle\lambda, f(x)\rangle - \frac{1}{N}\sum_{j\neq k}^{N}\langle\lambda, f(x)\rangle + \langle\lambda, f(x_k')\rangle||$$

$$\leq \sup_{\lambda\in\Omega_{\mathcal{S}}, f\in\mathcal{F}} \frac{1}{N} |\langle\lambda, f(x_k) - f(x_k')\rangle|$$

$$\leq \frac{2}{N}\sup_{\lambda\in\Omega_{\mathcal{S}}, f\in\mathcal{F}} ||\lambda||_1 ||f||_\infty = \frac{C}{N} \quad (C = 2\sup_{\lambda\in\Omega_{\mathcal{S}}, f\in\mathcal{F}} ||\lambda||_1 ||f||_\infty)$$

Now, by Mc Diarmid's inequality, by studying the function concerning its sampled expectation $\mathbb{E}_{\tilde{\mathcal{X}}}h(\cdot)$ over the samples set $\tilde{\mathcal{X}} = \{x_1, \ldots, x_N\}$:

$$P(h(x_1, \ldots, x_N) - \mathbb{E}_{\tilde{\mathcal{X}}}h(x_1, \ldots, x_k', \ldots, x_N) \geq \epsilon) \leq \exp(\frac{-2N\epsilon^2}{C^2})$$

$$P\left(h(x_1, \ldots, x_N) - \mathbb{E}_{\tilde{\mathcal{X}}}h(x_1, \ldots, x_k', \ldots, x_N) \geq C\sqrt{\frac{\log 1/\delta}{2N}}\right) \leq \delta$$

It then follows that

$$\max_{\eta \in \mathcal{S}} |\mathbb{E}_{\eta^\pi}[\log \eta] - \frac{1}{N} \sum_{j=0}^{N} \log \eta(x_j)| \leq \mathbb{E}_{\tilde{\mathcal{X}}} \sup_{\lambda \in \Omega_{\mathcal{S}}, f \in \mathcal{F}} |\mathbb{E}_{\eta^\pi} \langle \lambda, f(x) \rangle - \frac{1}{N} \sum_{j=0}^{N} \langle \lambda, f(x) \rangle| + C\sqrt{\frac{\log 1/\delta}{2N}}$$

We now use symmetrization techniques by considering the Rademacher sequence $\{\omega_j\}$ and by using the standard result that given a class of measurable functions $\mathcal{G}$ if

$$Z(\tilde{\mathcal{X}}) = \sup_{g \in \mathcal{G}} |\mathbb{E}g(x) - \frac{1}{N} \sum_{j=0}^{N} g(x_j)| \quad \text{and } R(\tilde{\mathcal{X}}, \omega) = \sup_{g \in \mathcal{G}} |\frac{1}{N} \sum_{j=0}^{N} \omega_j g(x_j)|$$

Then:

$$\mathbb{E}_{\tilde{\mathcal{X}}} Z(\tilde{\mathcal{X}}) \leq 2\mathbb{E}_{\tilde{\mathcal{X}}, \omega} R(\tilde{\mathcal{X}})$$

From this, it follows that the whole expression reduces to

$$\max_{\eta \in \mathcal{S}} |\mathbb{E}_{\eta^\pi}[\log \eta] - \frac{1}{N} \sum_{j=0}^{N} \log \eta(x_j)| \leq 2\mathbb{E}_{\tilde{\mathcal{X}}, \omega} \sup_{\lambda \in \Omega_{\mathcal{S}}, f \in \mathcal{F}} |\frac{1}{N} \sum_{j=0}^{N} \omega_j \langle \lambda, f(x_j) \rangle| + C\sqrt{\frac{\log 1/\delta}{2N}}$$

We extract the supremum over $\lambda \in \Omega_{\mathcal{S}}$ to obtain the (absolute) Rademacher averages of the functions in $\mathcal{F}$

$$\mathbb{E}_\omega \sup_{\lambda \in \Omega_{\mathcal{S}}, f \in \mathcal{F}} |\frac{1}{N} \sum_{j=0}^{N} \omega_j \langle \lambda, f(x_j) \rangle| \leq \sup_{\lambda \in \Omega_{\mathcal{S}}} ||\lambda||_1 \mathbb{E}_\omega \sup_{f \in \mathcal{F}} |\frac{1}{N} \sum_{j=0}^{N} \omega_j f(x_j)|$$

$$\leq \sup_{\lambda \in \Omega_{\mathcal{S}}} ||\lambda||_1 \mathcal{R}_N(\Phi)$$

It follows that the final formulation for the term we are studying is the following

$$\max_{\eta \in \mathcal{S}} |\mathbb{E}_{\eta^\pi}[\log \eta] - \frac{1}{N} \sum_{j=0}^{N} \log \eta(x_j)| \leq 2 \sup_{\lambda \in \Omega_{\mathcal{S}}} ||\lambda||_1 \mathcal{R}_N(\Phi) + C\sqrt{\frac{\log 1/\delta}{2N}}$$

$$C = 2 \sup_{\lambda \in \Omega_{\mathcal{S}}, f \in \mathcal{F}} ||\lambda||_1 ||f||_\infty$$

$\square$

**Final Step**

The bound offered by Lemma 6 would be unpractical since it relates a quantity central to our analysis to something which is not known in advance. Due to this, we make a further effort with the following Lemma, by substituting the term $\sup_{\lambda \in \Omega_{\mathcal{S}}} ||\lambda||_1$ with $||\hat{\lambda}||_1$. To do this, an additional assumption over the feature functions will be needed though. First of all, we bound the two terms in $\Omega_{\mathcal{S}}$ with

**Lemma 7.** *The solutions of the expected and sampled Max-Ent problem are related to the bound:*

$$||\bar{\lambda}||_1 \leq ||\hat{\lambda}||_1 + \sqrt{\frac{6M}{\sigma_{\min}(\hat{\text{Cov}}(\mathcal{F}))} \max_{\eta \in \mathcal{S}} |\mathbb{E}_{\eta^\pi}[\log \eta] - \frac{1}{N} \sum_{j=0}^{N} \log \eta(x_j)|}$$

*Proof.* We take into account the following relationships which are valid for the solutions of the MaxEnt problem under structural constraints, i.e. $\mathbb{E}_{\bar{\eta}}[f] = \mathbb{E}_{\eta^\pi}[f]$ and $\mathbb{E}_{\hat{\eta}}[f] = \mathbb{E}_{\tilde{\eta}}[f]$

$$H(\eta) = \log \sum_y \exp(\langle \lambda, f(y) \rangle) - \langle \lambda, \mathbb{E}_\eta[f] \rangle$$

$$= A(\lambda) - \langle \lambda, \mathbb{E}_\eta[f] \rangle = A(\lambda) - \langle \lambda, \nabla A(\lambda) \rangle$$

From which it follows that it is possible to recover the Bregman divergence under the log-partition function $D_A(\lambda_1, \lambda_2)$

$$
\begin{aligned}
H(\bar{\eta}) - H(\hat{\eta}) &= A(\bar{\lambda}) - A(\hat{\lambda}) - \langle \bar{\lambda}, \nabla A(\bar{\lambda}) \rangle + \langle \hat{\lambda}, \nabla A(\hat{\eta}) \rangle \\
&= A(\bar{\lambda}) - A(\hat{\lambda}) - \langle \bar{\lambda}, \nabla A(\bar{\lambda}) \rangle + \langle \hat{\lambda}, \nabla A(\hat{\lambda}) \rangle + \langle \bar{\lambda}, \nabla A(\hat{\lambda}) \rangle - \langle \bar{\lambda}, \nabla A(\hat{\eta}) \rangle \\
&= A(\bar{\lambda}) - A(\hat{\lambda}) - \langle \bar{\lambda} - \hat{\lambda}, \nabla A(\hat{\lambda}) \rangle + \langle \hat{\lambda}, \nabla A(\hat{\lambda}) - \nabla A(\bar{\lambda}) \rangle \\
&= D_A(\bar{\lambda}, \hat{\lambda}) + \langle \bar{\lambda}, \nabla A(\hat{\lambda}) - \nabla A(\bar{\lambda}) \rangle
\end{aligned}
$$

Now using the Taylor expansion of the divergence and the fact that $\nabla^2 A(\hat{\lambda}) = \hat{\mathrm{Cov}}(\mathcal{F})$

$$
\begin{aligned}
H(\bar{\eta}) - H(\hat{\eta}) + \langle \bar{\lambda}, \nabla A(\bar{\lambda}) - \nabla A(\hat{\lambda}) \rangle = D_A(\bar{\lambda}, \hat{\lambda}) \\
\geq \frac{1}{2}(\bar{\lambda} - \hat{\lambda})^\intercal \nabla^2 A(\hat{\lambda})(\bar{\lambda} - \hat{\lambda}) = \frac{1}{2}\|\bar{\lambda} - \hat{\lambda}\|^2_{\nabla^2 A(\hat{\lambda})} \\
\geq \sigma_{\min}(\nabla^2 A(\hat{\lambda}))\|\bar{\lambda} - \hat{\lambda}\|^2_2 \\
\geq \frac{\sigma_{\min}(\nabla^2 A(\hat{\lambda}))}{M}\|\bar{\lambda} - \hat{\lambda}\|^2_1 \\
\geq \frac{\sigma_{\min}(\hat{\mathrm{Cov}}(\mathcal{F}))}{M}\|\bar{\lambda} - \hat{\lambda}\|^2_1
\end{aligned}
$$

where $M$ corresponds to the number of the features. Finally, by exploiting the zero duality gap and the results of Lemma 10

$$
\begin{aligned}
\|\bar{\lambda} - \hat{\lambda}\|^2_1 &\leq \frac{M}{\sigma_{\min}(\mathrm{Cov}_{\hat{\lambda}}(f))}(H(\bar{\eta}) - H(\hat{\eta}) + \langle \bar{\lambda}, \nabla A(\bar{\lambda}) - \nabla A(\hat{\lambda}) \rangle) \\
&= \frac{M}{\sigma_{\min}(\hat{\mathrm{Cov}}(\mathcal{F}))}(\mathcal{L}_0(\bar{\lambda}) - \tilde{\mathcal{L}}(\hat{\lambda}) + \langle \bar{\lambda}, \nabla A(\bar{\lambda}) - \nabla A(\hat{\lambda}) \rangle) \\
&\leq \frac{M}{\sigma_{\min}(\hat{\mathrm{Cov}}(\mathcal{F}))}(|\mathcal{L}_0(\bar{\lambda}) - \tilde{\mathcal{L}}(\hat{\lambda})| + |\langle \bar{\lambda}, \nabla A(\bar{\lambda}) - \nabla A(\hat{\lambda}) \rangle|) \\
&\leq \frac{2M}{\sigma_{\min}(\hat{\mathrm{Cov}}(\mathcal{F}))}|\mathcal{L}_0(\bar{\lambda}) - \tilde{\mathcal{L}}(\hat{\lambda})| \\
&\leq \frac{6M}{\sigma_{\min}(\hat{\mathrm{Cov}}(\mathcal{F}))} \max_{\eta \in \mathcal{S}} |\mathbb{E}_{\eta^\pi}[\log \eta] - \frac{1}{N}\sum_{j=0}^{N} \log \eta(x_j)|
\end{aligned}
$$

It is then possible to write

$$
\|\bar{\lambda} - \hat{\lambda}\|_1 \leq \sqrt{\frac{6M}{\sigma_{\min}(\hat{\mathrm{Cov}}(\mathcal{F}))} \max_{\eta \in \mathcal{S}} |\mathbb{E}_{\eta^\pi}[\log \eta] - \frac{1}{N}\sum_{j=0}^{N} \log \eta(x_j)|}
$$

$$
|\|\bar{\lambda}\|_1 - \|\hat{\lambda}\|_1| \leq \|\bar{\lambda} - \hat{\lambda}\|_1 \leq \sqrt{\frac{6M}{\sigma_{\min}(\hat{\mathrm{Cov}}(\mathcal{F}))} \max_{\eta \in \mathcal{S}} |\mathbb{E}_{\eta^\pi}[\log \eta] - \frac{1}{N}\sum_{j=0}^{N} \log \eta(x_j)|}
$$

which concludes the proof. $\qquad\square$

Now, it is possible to combine all the previous results in

**Lemma 8.** *Assume that the minimum singular value of the sampled covariance matrix is strictly positive, that is $\sigma_{\min}(\hat{\mathrm{Cov}}(\mathcal{F})) > 0$, then the supremum term of Lemma 6 can be bounded with*

$$
\max_{\eta \in \mathcal{S}} |\mathbb{E}_{\eta^\pi}[\log \eta] - \frac{1}{N}\sum_{j=0}^{N} \log \eta(x_j)| \precsim 2\|\hat{\lambda}\|_1 \mathcal{R}_N(\Phi) + 2\|\hat{\lambda}\|_1 F\sqrt{\frac{\log 1/\delta}{2N}}
$$

*Proof.* Taking all together the terms obtained so far from Lemmas [6, 7], setting $C = 2\sup_{\lambda \in \{\bar{\lambda}, \hat{\lambda}\}, f \in \mathcal{F}} ||\lambda||_1 ||f||_\infty$ we have

$$\max_{\eta \in \mathcal{S}} |\mathbb{E}_{\eta^\pi}[\log \eta] - \frac{1}{N} \sum_{j=0}^{N} \log \eta(x_j)| \leq 2 \sup_{\lambda \in \{\bar{\lambda}, \hat{\lambda}\}} ||\lambda||_1 \mathcal{R}_N(\Phi) + C\sqrt{\frac{\log 1/\delta}{2N}}$$

$$\sup_{\lambda \in \{\bar{\lambda}, \hat{\lambda}\}} ||\lambda||_1 \leq ||\hat{\lambda}||_1 + \sqrt{\frac{6M}{\sigma_{\min}(\hat{\text{Cov}}(\mathcal{F}))} \max_{\eta \in \mathcal{S}} |\mathbb{E}_{\eta^\pi}[\log \eta] - \frac{1}{N} \sum_{j=0}^{N} \log \eta(x_j)|}$$

It follows the quadratic form in $x = \sqrt{\max_{\eta \in \mathcal{S}} |\mathbb{E}_{\eta^\pi}[\log \eta] - \frac{1}{N} \sum_{j=0}^{N} \log \eta(x_j)|}$

$$x^2 - bx - c \leq 0$$

$$b = 2\sqrt{\frac{6M}{\sigma_{\min}(\hat{\text{Cov}}(\mathcal{F}))}} \left[ \mathcal{R}_N(\Phi) + F\sqrt{\frac{\log 1/\delta}{2N}} \right] \geq 0$$

$$c = 2||\hat{\lambda}||_1 \left[ \mathcal{R}_N(\Phi) + F\sqrt{\frac{\log 1/\delta}{2N}} \right] \geq 0$$

The discriminant is well defined $\Delta = b^2 + 4c \geq 0$ and the solution is given by

$$\max_{\eta \in \mathcal{S}} |\mathbb{E}_{\eta^\pi}[\log \eta] - \frac{1}{N} \sum_{j=0}^{N} \log \eta(x_j)| \leq \left( \frac{b + \sqrt{b^2 + 4c}}{2} \right)^2$$

$$\leq \frac{b^2}{2} + c + b\sqrt{b^2 + 4c}$$

$$\precsim 2||\hat{\lambda}||_1 \mathcal{R}_N(\Phi) + 2||\hat{\lambda}||_1 F\sqrt{\frac{\log 1/\delta}{2N}}$$

The final step was done because all additional terms out of $c$ itself are of higher order. □

## B Further instrumental Lemmas

In this section, we present some additional standard lemmas which summarize some important properties of the Max-Ent solutions and distributions in the exponential family that was used in the employed section.

**Lemma 9.** *For any distribution $\eta$ in the exponential family, it holds that for the log-likelihood with respect to a distribution $\eta^\pi$ it holds that*

$$\mathcal{L}_{\eta^\pi}(\lambda) = A(\lambda) - \langle \lambda, \mathbb{E}_{\eta^\pi}[f] \rangle$$

*Proof.*

$$\mathcal{L}_{\eta^\pi}(\lambda) = -\mathbb{E}_{\eta^\pi}[\log \eta] = -\mathbb{E}_{\eta^\pi}[\langle \lambda, f \rangle - \log \Phi_\lambda] = -\langle \lambda, \mathbb{E}_{\eta^\pi}[f] \rangle + A(\lambda)$$

□

**Lemma 10.** *For any distribution $\eta$ in the exponential family, it holds that*

$$|\mathcal{L}_{\eta^\pi}(\lambda) - \tilde{\mathcal{L}}(\lambda)| = |\langle \lambda, \mathbb{E}_{\eta^\pi}[f] - \tilde{\mathbb{E}}[f] \rangle|$$

*where $\mathcal{L}_{\eta^\pi}(\lambda)$ is the negative log-likelihood of $\eta$ with respect to $\eta^\pi$.*

*Proof.*

$$|\mathcal{L}_{\eta^\pi}(\lambda) - \tilde{\mathcal{L}}(\lambda)| = |-\langle \lambda, \mathbb{E}_{\eta^\pi}[f] \rangle + A(\lambda) + \langle \lambda, \mathbb{E}_{\tilde{\eta}}[f] \rangle - A(\lambda)|$$
$$= |\langle \lambda, -\mathbb{E}_{\eta^\pi}[f] + \mathbb{E}_{\tilde{\eta}}[f] \rangle|$$
$$= |\langle \lambda, \mathbb{E}_{\eta^\pi}[f] - \mathbb{E}_{\tilde{\eta}}[f] \rangle|$$

□

We will now derive some properties between the sampled log-likelihood and the log-likelihood with respect to the true distribution $\eta^\pi$, called $\mathcal{L}_0$ for simplicity

**Lemma 11.** *For the solutions of the exact and sampled Max-Ent problems, $\bar{\eta}$ and $\hat{\eta}$ respectively, it holds that*

$$|\mathcal{L}_0(\bar{\lambda}) - \tilde{\mathcal{L}}(\hat{\lambda})| \leq |\langle \hat{\lambda}, \mathbb{E}_{\eta^\pi}[f] - \mathbb{E}_{\tilde{\eta}}[f]\rangle|$$
$$\leq |\langle \hat{\lambda}, \nabla A(\bar{\lambda}) - \nabla A(\hat{\lambda})\rangle|$$

$$|\mathcal{L}_0(\bar{\lambda}) - \tilde{\mathcal{L}}(\hat{\eta})| \geq |\langle \bar{\lambda}, \mathbb{E}_{\eta^\pi}[f] - \mathbb{E}_{\tilde{\eta}}[f]\rangle|$$
$$\geq |\langle \bar{\lambda}, \nabla A(\bar{\lambda}) - \nabla A(\hat{\lambda})\rangle|$$

*Proof.* The proof follows directly from the fact that $\bar{\lambda}$ is optimal with respect to $\hat{\eta}$ in the exact problem $\mathcal{L}_0(\bar{\lambda}) \leq \mathcal{L}_0(\hat{\lambda})$ and viceversa $\tilde{\mathcal{L}}(\bar{\lambda}) \geq \tilde{\mathcal{L}}(\hat{\eta})$. $\quad\square$

## C   Monotonicity Lemma

In this section, we provide the proof of Lemma 2.

*Proof.* Taking into account two features with increased factorization $\mathcal{F} \subset \mathcal{F}'$ we consider the particular set of factorized features $\bar{\alpha}, \{\bar{\alpha}_k\}$, since the rest of the features are the same. It follows that

$$|\hat{\mu}_{\bar{\alpha}}| = \sum_k |\hat{\mu}_{\bar{\alpha}_k}|$$
$$\frac{|\hat{\mu}_{\bar{\alpha}}|}{|S_{\bar{\alpha}}|} = \sum_k \frac{|\hat{\mu}_{\bar{\alpha}_k}|}{|S_{\bar{\alpha}}|} \quad (\forall \bar{\alpha}_k : |S_{\bar{\alpha}_k}| < |S_{\bar{\alpha}}|)$$
$$\frac{|\hat{\mu}_{\bar{\alpha}}|}{|S_{\bar{\alpha}}|} \leq \sum_k \frac{|\hat{\mu}_{\bar{\alpha}_k}|}{|S_{\bar{\alpha}_k}|}$$

Now, due to the relationship of Lemma 12 we know that $\hat{\lambda}_\alpha = f(\frac{|\hat{\mu}_{\bar{\alpha}}|}{|S_{\bar{\alpha}}|})$ with $f(\cdot)$ being an unknown but subadditive for positive values of $\lambda$. Moreover, the functions are the same for all the terms, so that

$$\frac{|\hat{\mu}_{\bar{\alpha}}|}{|S_{\bar{\alpha}}|} \leq \sum_k \frac{|\hat{\mu}_{\bar{\alpha}_k}|}{|S_{\bar{\alpha}_k}|}$$
$$f(\frac{|\hat{\mu}_{\bar{\alpha}}|}{|S_{\bar{\alpha}}|}) \leq f(\sum_k \frac{|\hat{\mu}_{\bar{\alpha}_k}|}{|S_{\bar{\alpha}_k}|}) \leq \sum_k f(\frac{|\hat{\mu}_{\bar{\alpha}_k}|}{|S_{\bar{\alpha}_k}|})$$
$$|\lambda_{\bar{\alpha}}| \leq \sum_k |\lambda_{\bar{\alpha}_k}|$$

Since the rest of the terms are the same, this concludes the proof. $\quad\square$

**Lemma 12.** *There exists a monotonic and anti-symmetric function $f(\cdot)$ such that it is possible to univocally define $\hat{\lambda}_\alpha = f(\hat{\mu}_\alpha, |S_\alpha|, G_{max})$*

*Proof.* We start by considering the Lagrangian formulation of the Max-Ent problem,

$$\mathcal{L}(\eta, \lambda) = H(\eta) + \sum_{\alpha \in \mathcal{I}_\mathcal{F}} \lambda_\alpha (\mathbb{E}_\eta[f_\alpha] - \hat{\mu}_\alpha) + \mu(\mathbb{E}[\eta] - 1) \tag{17}$$

By taking the gradient of the Lagrangian with respect to the distribution it follows that each $x$-term of the support gives

$$(\nabla_\eta \mathcal{L})(x) = -1 - \log \eta(x) + \lambda_\alpha f_\alpha + \mu$$

From which it follows that with $\lambda_0 = \mu - 1$ the equation for the $\alpha$-constraint is

$$\eta_\alpha(x) = e^{\lambda_0} e^{\lambda_\alpha f_\alpha(x)}$$

We now compute insert this value inside the constraint equation under the feature class $f_\alpha = g \mathbb{1}_{s \in \mathcal{S}_\alpha}$

$$\int_\mathcal{R} \int_\mathcal{X} g\eta(x)_\alpha = \hat{\mu}_\alpha$$

$$|S_\alpha| \int_{G_{\min}}^{G_{\max}} g e^{\lambda_0} e^{\lambda_\alpha r} dg = \hat{\mu}_\alpha$$

which leads to the implicit formulation for $\lambda_\alpha$ by solving the integral by setting $G = G_{\max}$

$$e^{\lambda_0} \frac{2\lambda_\alpha \cosh(G\lambda_\alpha) - 2\sinh(G\lambda_\alpha))}{\lambda_\alpha^2} = \frac{\hat{\mu}_\alpha}{|S_\alpha|}$$

Now, it can be proven by considering the normalization constraint that $e^{\lambda_0} = 1/Z(\lambda)$ with $Z(\lambda)$ a constant depending on $\lambda_\alpha$, in particular:

$$Z(\lambda) = \int_\mathcal{X} e^{\sum_\alpha \lambda_\alpha f_\alpha} dx$$

$$= \sum_\alpha |S_\alpha| \int_\mathcal{R} e^{\lambda_\alpha f_\alpha} dr$$

$$= \sum_\alpha |S_\alpha| \frac{\sinh \lambda_\alpha G}{\lambda_\alpha}$$

$$= |S_\alpha| \frac{\sinh \lambda_\alpha G}{\lambda_\alpha} + C$$

The whole equation then becomes

$$\frac{2\lambda_\alpha \cosh(G\lambda_\alpha) - 2\sinh(G\lambda_\alpha))}{\lambda_\alpha^2} = \hat{\mu}_\alpha \left(\frac{\sinh \lambda_\alpha G}{\lambda_\alpha} + C\right)$$

This equation provides an implicit definition for $\lambda_\alpha$ and it can be shown to be convex for positive values of $\lambda$. The function for lambda is the inverse of this whole term, which is then concave and has a zero in the origin, thus it is sub-additive. $\qquad\square$

## D   Further Discussion about the Environment Setup

The GridWorld instance was selected to be able to satisfy the Experiment Objectives. More specifically, for the following reasons:

- be able to test the representation learning capacity of the algorithm in a setting with an apparent factorization of the state space
- because of the possibility of explicitly comparing the outputs with the true distribution.

As previously said, the simulations were run on a rectangular GridWorld, with a height of $4$ and length of $8$, with traps on the whole second line and goals all over the top. A visualization of the Gridworld can be seen in Fig.5.

## E   Reproducibility

The code was run over a 4 core Intel(R) Core(TM) i5-4570 CPU @ 3.20GHz. The package use to perform Maximum Entropy density estimation can be found at Max-Ent. The repository can be found at the link Code.



Figure 5: GridWorld template