# OpenReview forum: "Distributional Policy Evaluation: a Maximum Entropy approach to Representation Learning"
_NeurIPS.cc/2023/Conference — NeurIPS 2023 poster_

### Official Review · Reviewer_wzKJ · 2023-07-01

**Soundness:** 1 poor
**Presentation:** 2 fair
**Contribution:** 1 poor
**Rating:** 2
**Confidence:** 4

**Summary:**

This paper investigates policy evaluation of distributional RL by leverage of Maximum entropy density estimations without explicitly considering Bellman operator and contraction mapping. They further derived the new generalization error bound in the maximum entropy PE process. For a practical algorithm, authors proposed progressive factorization to refine the representation with desirable properties, including the monotonicity. They did some experiments to demonstrate their claims.



**Strengths:**

* Introducing Maximum entropy density estimate into distributional RL has not been studied.
* Progressive factorization seems to refine the representation of RL.
* The writing is basically clear.


**Weaknesses:**

* The motivation is not clear. Why should we consider the maximum entropy estimate to study the representation?
* Without discussing the RL context in the proposal, e.g., Bellman operator and contraction mapping, the technical soundness of the so-called max-ent approach is questionable.
* Experiments are weak and not directly related to the distributional RL setting.


**Questions:**

To the best of our knowledge, the representation issue in (distributional) RL is not well-defined yet. Without discussing what the representation issue you are going to study, directly incorporating an approach is very confusing to me. Also, there are three major issues I would raise here.

**1.Motivation.** What is the representation issue in distributional RL? Is it just the representation of the value distribution? Note that people may also think distributional RL is also able to improve the representation of environments by additionally leverage of return distribution information. To be honest, after carefully the whole paper, I am still confused why your proposed methods are beneficial for representation. If authors refer to the monotonicity properties of progressive factorization, are there any experiments on real environments, e.g., Atari games, to show these benefits?

**2.Less discussion within RL context**. The authors claim the proposed method can handle the representation issue in RL context, i.e., policy evaluation, but they did not rigorously prove the contraction properties of Bellman operator under the maximum entropy density estimate. Note that as an RL research paper, the discussion of contraction properties is the foundation of variants of policy evaluation algorithms. Otherwise, it would be technically problematic. In line 49, the authors mentioned PE in a distributional setting is directly linked to the representation issue, but as far as I can tell, they are not equivalent. Discussing the contraction mapping properties is necessary.

**3.Experiments are weak**. If authors claim they are handling the representation issue of distributional RL, I do not find any direct experiment to demonstrate it on commonly used environments.  Applying the state-aggregation approach in practice is also problematic as it hinders the generalization for RL with function approximation.  In line 128, the authors claimed that Max-ent has multiple advantages. Have authors demonstrated them in their experiments? Similar issues also apply to the monotonicity property in Line 205.


Overall, I do not think this paper really handles the representation issue in the real distributional RL setting. Some preliminary results are given without connection with Bellman equation and RL context. Experiments are also weak to support their ambitious targets mentioned in the introduction.

---

> ### Author Rebuttal · Authors · 2023-08-09
>
> We thank the reviewer for the thorough review.  We will answer proceeding by points:
>
> *What is the representation issue in distributional RL? Is it just the representation of the value distribution? Note that people may also think distributional RL is also able to improve the representation of environments by additionally leverage of return distribution information. To be honest, after carefully the whole paper, I am still confused why your proposed methods are beneficial for representation. If authors refer to the monotonicity properties of progressive factorization, are there any experiments on real environments, e.g., Atari games, to show these benefits?*
>
> As for the high-level representation-learning definition, we referred to it rather informally mostly because the representation issue is in general not well-defined, but we will for sure add an explicit description of what we mean by representation-learning in this context:  it consists of **finding a good feature representation of the decision-making space so as to make the overall learning problem easier**. In this sense, **a method able to reduce the dimensionality of the problem is said to address the representation-learning problem.** In this sense, the representation-learning issue is the same in distributional RL as in expected-value RL, and canonical (distributional) RL methods indirectly address it by means of generic function approximation. Nonetheless, **distributional methods further exacerbate this problem with respect to expected-value RL**, since they require learning distributions over (usually) large spaces. For this reason, we believe that investigating how to use distributions of returns to alleviate the issue is an interesting stream of research.
>
> *The authors claim the proposed method can handle the representation issue in RL context, i.e., policy evaluation, but they did not rigorously prove the contraction properties of Bellman operator under the maximum entropy density estimate. Note that as an RL research paper, the discussion of contraction properties is the foundation of variants of policy evaluation algorithms. Otherwise, it would be technically problematic. In line 49, the authors mentioned PE in a distributional setting is directly linked to the representation issue, but as far as I can tell, they are not equivalent. Discussing the contraction mapping properties is necessary.*
>
> Up to our knowledge, contraction considerations are indeed a fundamental tool in the RL methods that enforce the MDP structure (i.e. TD-based), as canonical distributional RL methods devised so far. On the other hand,  **contraction considerations are meaningless for Monte-Carlo-based methods, since no operator is actually defined**; our work is indeed in this stream of work, and **this motivates the absence of contraction considerations in the analysis**. We will for sure add one comment giving reasons for the absence of contraction arguments, but more generally we do not agree with the claim that "Discussing the contraction mapping properties is necessary", we rather believe that  "**Discussing the contraction mapping properties is necessary for operator-based methods**", which is not the case.  We point out that a large portion of sound and effective RL algorithms (including policy gradients) are based on Monte Carlo simulations and do not make use of Bellman operators. While we agree that Mote-Carlo methods might overlook some of the MDP structure, they have the advantage of leading to (i) a simpler mathematical treatment and (ii) of being applicable even to non-Markovian environments.
> As for the final part of the question, the aim of the paper was indeed to show that policy evaluation and representation learning were linked, and in fact, **we showed that performing PE can drive the learning of a reduced-dimension representation**. Yet, **the fact that they were equivalent was not among the objectives of our work**. We can further specify this if the reviewer thinks some parts of the exposition were misleading.
>
> *If authors claim they are handling the representation issue of distributional RL, I do not find any direct experiment to demonstrate it on commonly used environments. Applying the state-aggregation approach in practice is also problematic as it hinders the generalization for RL with function approximation. In line 128, the authors claimed that Max-ent has multiple advantages. Have authors demonstrated them in their experiments? Similar issues also apply to the monotonicity property in Line 205.*
>
> The main contribution of the paper is of theoretical nature. For this reason, the purpose of the simulations was not to demonstrate empirically the effectiveness of the method, but  to illustrate two essential features of the proposed method that were only suggested by the theoretical result, so as to close all the topics directly or indirectly introduced in the theoretical study:
> - whether it is possible to boost factorization by tuning the value of $\beta$
> - whether the resulting factorization aggregated similar states in terms of the policy's true return distribution;
>
> The MDP instances we employed were designed to be coherent with the objectives of the simulations and to be as much interpretable as possible in terms of true return distributions, a feature which is hardly shared with more complex RL tasks (e.g., Atari). We believe that the adaptation of the proposed method to more practical contexts is out of the scope of the present paper.

---

> > ### Comment · Reviewer_wzKJ · 2023-08-15
> > **Thank you for Response**
> >
> > Firstly, I thank the authors for their detailed responses. However, after reading the other reviewers' responses as well as the authors' rebuttal, I am afraid I still hold divergent opinions from the authors on some of the key issues of this paper.
> >
> > * If the authors claimed that they are targeting representation learning by reducing the dimension of state features, why does this paper not consider real RL environments, like Atari games, where the state representation is critical enough? Although the authors claimed this is a theory paper, I do not think lacking important experiments on real envs will hit the standard of this venue.
> >
> > * For the theory contribution that the authors emphasized, I have been working on distributional RL for many years, and I do not think an analysis of KL divergence is directly related to typical distributional RL algorithms, including QRDQN, IQN, MMDRL. I noticed the discussion of authors in lines 302 to 311, which is actually critical in the practical distributional RL. Since the title of this is distributional policy evaluation, it will be misleading as a similar concept has already been investigated in distributional RL with an explicit distributional Bellman operator defined[1, 2]. That is why I suggest the contraction mapping of the Bellman operator should be important.
> >
> > * Based on my expertise in RL, MC-based algorithms are the sample-based extension of model-based Dynamical Programming (DP) algorithms. Since vanilla Policy evaluation is clearly a DP version, it is natural to expect some properties, like the contraction, in DP, should be explicitly defined and discussed.
> >
> > * Another question is whether a direct analysis of the generalization error bound is valid in RL theory paper, where policy evaluation is very different from supervised learning. Although there indeed exist some links, I am afraid this point has not been clearly stated in this paper. Based on these reasons, I do not think it is a solid theory paper, either,  for the venue with high standards.
> >
> >
> > In summary, I am afraid at this point I need to keep my score highly based on my first impression of this paper, although my evaluation is divergent from the others. I am also open to any discussion with the other reviewers and AC in the following period.
> >
> >
> >
> > [1] DSAC: Distributional Soft Actor Critic for Risk-Sensitive Reinforcement Learning
> >
> > [2] Interpreting Distributional Reinforcement Learning: A Regularization Perspective

---

> > > ### Author Response · Authors · 2023-08-16
> > >
> > > We thank the reviewer for the reply.
> > >
> > > *If the authors claimed that they are targeting representation learning by reducing the dimension of state features, why does this paper not consider real RL environments, like Atari games, where the state representation is critical enough? Although the authors claimed this is a theory paper, I do not think lacking important experiments on real envs will hit the standard of this venue.*
> > >
> > > We believe that, for a theory paper, illustrative RL domains, such as the ones we provide, are more appropriate than applications to complex environments. Indeed, illustrative domains are able to clearly highlight the features of our approach, while in more complex environments the effect of our representation learning approach could be hidden behind other confounding effects (e.g., difficulty in the training of deep neural networks, choice of the proper architecture, overfitting). We also point out that the other reviewers suggested that further experiments would have been appreciated, but they believed that the contribution is still sufficient for acceptance.
> > >
> > > *For the theory contribution that the authors emphasized, I have been working on distributional RL for many years, and I do not think an analysis of KL divergence is directly related to typical distributional RL algorithms, including QRDQN, IQN, MMDRL. I noticed the discussion of authors in lines 302 to 311, which is actually critical in the practical distributional RL. Since the title of this is distributional policy evaluation, it will be misleading as a similar concept has already been investigated in distributional RL with an explicit distributional Bellman operator defined[1, 2]. That is why I suggest the contraction mapping of the Bellman operator should be important.*
> > >
> > > As the Reviewer noticed, **in the "Related Works" we highlighted that "Our work tries to answer different research questions compared to traditional policy evaluation in D-RL"**. Additionally, we highlighted the fact that KL-divergence is not related to other standard D-RL quantities without further assumptions (311) and that we apply completely different techniques (304-305). MC-approaches (which are alternatives to TD methods and do not make use of the Bellman operator) are qualified methods for policy evaluation (see also answer to the next point). Finally, we take the liberty to point out that **the fact that this work is different in many ways with respect to other D-RL works and does not follow the mainstream research in D-RL should be regarded as an element of novelty, rather than an issue**.

---

> > > ### Author Response · Authors · 2023-08-16
> > >
> > >
> > > *Based on my expertise in RL, MC-based algorithms are the sample-based extension of model-based Dynamical Programming (DP) algorithms. Since vanilla Policy evaluation is clearly a DP version, it is natural to expect some properties, like the contraction, in DP, should be explicitly defined and discussed.*
> > >
> > > **MC-based algorithms** for policy evaluation have been extensively used both in the expected RL and D-RL communities. They **are not "sample-based extensions of model-based Dynamical Programming (DP) algorithms"** since they make no use of the Bellman operator and, consequently, they have never been studied in terms of contraction. In particular, the chapter "Monte-Carlo prediction" of Sutton and Barto's book [1] states "Whereas the DP diagram includes only one-step transitions, the Monte Carlo diagram goes all the way to the end of the episode. These differences in the diagrams accurately reflect the fundamental differences between the algorithms. [...] **In other words, Monte Carlo methods do not bootstrap**". The same phrasing is reported in the chapter "Learning the Return Distribution" of Bellemare, Dabney, and Rowland's book [2], and here as well no contraction properties of distributional Monte-Carlo vanilla Policy Evaluation are shown.
> > >
> > > [1]- Reinforcement Learning: An Introduction, Richard S. Sutton, Andrew G. Barto (2022)
> > >
> > > [2]- Distributional Reinforcement Learning, Marc G. Bellemare and Will Dabney and Mark Rowland (2023)
> > >
> > >
> > > *Another question is whether a direct analysis of the generalization error bound is valid in RL theory paper, where policy evaluation is very different from supervised learning. Although there indeed exist some links, I am afraid this point has not been clearly stated in this paper. Based on these reasons, I do not think it is a solid theory paper, either, for the venue with high standards.*
> > >
> > > We believe to have indeed treated the link at the very beginning of Section 3: "The proposed approach turns distributional PE into a pure density estimation problem" (116-118), yet "[to] have access to a batch of i.i.d. samples, this is not necessarily restrictive: the result can be generalized for a single $\beta$-mixing sample path by exploiting blocking techniques" (135-138). It is common in the RL literature to derive theoretical results for the i.i.d. setting (like in supervised learning) since they can be comfortably converted into results for dependent samples under the $\beta$-mixing of the underlying chain [3,4]. Thus, **the fact that it is possible to formalize a PE problem so as to derive generalization error bounds when MC estimates are employed is indeed a valid result for RL and a part of the novel contributions of this work.**
> > >
> > > [3]- Rates of Convergence for Empirical Processes of Stationary Mixing Sequences, Bin Yu (1994)
> > >
> > > [4]- DualDICE: Behavior-Agnostic Estimation of Discounted Stationary Distribution Corrections, Ofir Nachum, Yinlam Chow, Bo Dai, and Lihong Li (2019)
> > >
> > >
> > > We hope that these answers might elucidate the remaining doubts and help the Reviewers discuss them in the following period.

---

### Official Review · Reviewer_SYd5 · 2023-07-03

**Soundness:** 3 good
**Presentation:** 3 good
**Contribution:** 3 good
**Rating:** 6
**Confidence:** 3

**Summary:**

This paper proposes two algorithms, distributional maximum entropy policy evaluation (D-Max-Ent PE) and D-Max-Ent Progressive Factorization.

The first algorithm applies the formulations of maximum entropy RL to the problem of distributional policy evaluation, while the second extends the first by adding a progressive state space factorization step that recursively decomposes/factorizes the state space by evaluating the maxent PE at each resolution to find a decomposition that minimizes the generalization error bound.


**Strengths:**

I found this paper to be generally well written and understandable. A few grammar issues are present, but nothing that seriously hurts legibility, and the math in the main paper flows intuitively.

Distributional maxent policy evaluation makes sense as an approach to PE. I'm not familiar with recent literature on PE methods, but it seems like a good approach to the problem, and maxent methods have had enough empirical success in deep RL work that it seems like a reasonable approach here too.

I thought combining maxent PE with an iterative state space decomposition to be a clever and intriguing idea as well. On paper it makes sense to me as a way to divide and conquer large state spaces, and I'm curious how this idea could extend to more complex tabular and continuous MDPs (as they authors note it should).

Overall this seems like an interesting theoretical approach with appealing generality and the potential for extension to more complex domains.

**Weaknesses:**

The main issue I found with this paper is the relative simplicity of the domain studied, especially in the limited empirical evaluation.

I recognize this is a theory paper, and many significant practical questions are yet to be addressed. In particular the problem of how to factorize a state space is very hard in general and a worthy topic in its own right (the authors do acknowledge this). However, I'd still like to see some more analysis of what sorts of MDPs the proposed algorithm is strong in versus weak using just the simple factorizations discussed here, ideally in the form of additional experiments on other gridworld MDPs besides the single example case. I think this idea is interesting, so as an experimentalist I'd like to see its practical behavior demonstrated a little more.

I'm still inclined to recommend acceptance as the core ideas seem well motivated and interesting, but I think this would be a stronger paper with some more validation.


**Questions:**

I don't see distributional policy evaluation defined explicitly anywhere (though I think it's mostly covered by equation 3 and surrounding material?), unlike maxent RL which is defined in detail. For clarity this seems like it would be nice to have.

I like the use of specific grounding questions in the introduction nice, but I found the wording of the questions to be kind of leading. They seem phrased in such a way that the fact of asking them means the answer will be "yes" which feels not very useful rhetorically. Perhaps they can be reworded to be more thought provoking?

Absent further experiments, it'd be nice to have some analysis and discussion of what regimes these algorithms will tend to perform well in versus struggle in. For example, what happens to the test case when the number of states increases, in X versus Y dimension? Are there particular properties of an MDP that will lead to high performance/better error bounds?

**Limitations:**

The core of the limitations of this work are covered in my discussion of weaknesses. The authors are pretty clear about the extent of the contributions and the paper is pretty clear with its claims, however, which is always nice to see.

I don't see any potential negative societal impact issues stemming from this work.

---

> ### Author Rebuttal · Authors · 2023-08-09
>
> We thank the reviewer for the thorough review and for the constructive comments.
>
> *I don't see distributional policy evaluation defined explicitly anywhere (though I think it's mostly covered by equation 3 and surrounding material?), unlike maxent RL which is defined in detail. For clarity this seems like it would be nice to have.*
>
> Distributional PE was indeed left implicitly defined, but a **brief definition of the PE problem will be for sure added.**
>
> *I like the use of specific grounding questions in the introduction nice, but I found the wording of the questions to be kind of leading. They seem phrased in such a way that the fact of asking them means the answer will be "yes" which feels not very useful rhetorically. Perhaps they can be reworded to be more thought provoking?*
>
> The questions were mostly used to drive the discussion toward the desired direction, but as you suggest a more provoking wording might help the rhetoric of the whole work. We would propose as a rewording something like:
> - Q1:  Does employing return distributions offer other tools out of the standard distributional RL methods?
> - Q2: How are representation learning and policy evaluation intertwined? Do distributional methods offer a new way to highlight and exploit this connection?
>
> *Absent further experiments, it'd be nice to have some analysis and discussion of what regimes these algorithms will tend to perform well in versus struggle in. For example, what happens to the test case when the number of states increases, in X versus Y dimension? Are there particular properties of an MDP that will lead to high performance/better error bounds?*
>
> This point is an extremely interesting point indeed; we started addressing it (261-263) and left as a future contribution (337-338).  We will for sure add a section in the Appendix better explaining these considerations:
>    - In the considered MDP instances, increasing X or Y would affect the results based on which kind of change they would introduce in the return distribution landscape. In case the change would be consistent with the current factorization (for example, adding over the X dimension with the same grid layout) the results would follow; in general, **as far as the added dimensions would maintain the same factorization structure in the return distribution, the results would follow.**
>    - A factorizability structure was introduced but not directly addressed. In general, similarly to what is done in "On the Complexity of Representation Learning in Contextual Linear Bandits", Tirinzoni, Pirotta, Lararic (2023),  we believe that **a condition of "realizability" of the state-aggregation feature class for $\eta^\pi$ would be required**, or at least in an approximate formulation. Anyway, for state-aggregation feature classes in particular, the **high-performance MDP instances the reviewer mentioned would be the ones where acting with one policy would induce approximately the same return distribution in many different states**, which are the ones that will end up being aggregated.

---

> > ### Comment · Reviewer_SYd5 · 2023-08-16
> > **Response to rebuttal**
> >
> > Thanks for the added insight!
> >
> > Re: the rhetorical questions, I find that questions which should be answered either "yes" or "no" don't really engage the reader much, since the fact of asking the question in context tends to tell you what the answer is. The goal is for the reader to think about the question for a minute to get them invested in the answer you're about to provide, so in my experience something open-ended or hard-to-answer (but where an educated reader might be able to make a hypothesis) works well.
> >
> > In that vein, I like your proposed revised Q2, but Q1 still seems like the answer will be "yes" even if one doesn't understand the paper, since if the answer wasn't "yes" you probably wouldn't have written a paper about it. I'd suggest something like "What tools can return distributions provide for distributional RL?" or similar as an alternative.
> >
> > There's more interesting questions to dig into on this topic for sure, and I think the existing content is a good starting point. I hope to see more work on this topic to build on this foundation in future papers.

---

> > > ### Author Response · Authors · 2023-08-18
> > >
> > > We thank the Reviewer for the important suggestion, we will for sure change the questions to this more open-ended questions.

---

### Official Review · Reviewer_HRB1 · 2023-07-05

**Soundness:** 3 good
**Presentation:** 3 good
**Contribution:** 3 good
**Rating:** 5
**Confidence:** 1

**Summary:**

The authors proposed a novel Maximum Entropy framework for policy evaluation in a distributional RL that can explicitly take into account the features used to represent the state space while evaluating a policy. Based on the framework, the authors developed  D-Max-Ent Progressive Factorization algorithm balancing the trade-off between bias and variance of the representation space.

**Strengths:**

Overall well written and well organized paper. The authors answered two interesting research questions of representation learning. The authors analysed the questions both theoritically and experimentally by proposing a novel maximum entropy framework for policy evaluation and a new algorithm based on the framework.

**Weaknesses:**

Experiment setting and results presentation are a bit hard to understand. I'm not very familiar with representation learning and distributional RL, so maybe this is standard but I find the figures hard to interpret, so as the importance of the contribution of the paper based on the experiments.

**Questions:**

I see the experiments are conducted on a Grid task, will the proposed algorithm also applicable on continuous environments? Will the algorithm help with performance, training stability, sample efficiency?
(Again, I'm not familiar with representation learning and distributional RL so let me know if my question is not the main concern in the field.)
Maybe add some legends and more explanations about the figures to help present the results and emphasize the contribution.

**Limitations:**

Experiments designed in the paper help to support some theoretical analysis of the proposed algorithm, but I'm not sure about the benefits/advantage of the algorithm in a wider range of tasks based on my understanding of the paper.

---

> ### Author Rebuttal · Authors · 2023-08-09
>
> We thank the reviewer for the points outlined, in particular the ones about the readability of the figures.
>
> *I see the experiments are conducted on a Grid task, will the proposed algorithm also applicable on continuous environments? Will the algorithm help with performance, training stability, sample efficiency?*
>
>
> Indeed, the **proposed method works with continuous spaces as well**, yet the factorization rule is potentially less intuitive since it needs to cut continuous spaces. As for your second question,  **reducing the input space dimension to the decision-making problem would help sample efficiency** for sure, once it assures that the new representation does not introduce bias, as the proposed method does. On the other hand,  how to combine the proposed framework with existing methods in order to boost performance was left for future work.
>
> *Maybe add some legends and more explanations about the figures to help present the results and emphasize the contribution.*
>
>
> Due to space limitations, the description of the experiments was limited to the core, as for the figures which were made to contain all the needed information, but **we intend to extend this whole section in the additional page of the Camera Ready version**.

---

> > ### Comment · Reviewer_HRB1 · 2023-08-16
> > **Response to rebuttal**
> >
> > Thank the authors for their response to my questions. I'll maintain the score.

---

### Official Review · Reviewer_W4d8 · 2023-07-08

**Soundness:** 4 excellent
**Presentation:** 3 good
**Contribution:** 3 good
**Rating:** 6
**Confidence:** 4

**Summary:**

This paper takes a maximum entropy based approach to learning the return distribution of a policy. This framework learns a distribution with maximum entropy subject to matching the expectation of a certain set of functions, referred to as the feature functions. They adapted a classical maximum entropy error bound to measure the approximation error of their learnt distribution to the true return distribution in the KL divergence. They propose a framework where the features are based on underlying state abstractions, and propose an algorithm which progressively learns a return distribution and underlying features. They then evaluate their algorithm in various small scale experiments.

**Strengths:**

- Using a maximum entropy approach to learning the return distribution is an interesting direction, and novel from the previous parametric approaches used.
- This paper takes a step towards studying the interaction of distributional RL and representation learning.
- The theoretical results are cleanly proved, with very clear and organized arguments.

**Weaknesses:**

- The paper takes an RL agnostic approach, and learns an approximate return distribution without taking into account the underlying RL decision problem (i.e. Bellman operators aren't used, nor reward nor transition information, etc). This seems contrary to the decision-aware RL direction, which broadly states that we should focus learning capacity on aspects which are important for the decision problem. Some guarantees that learning a model in this way comes with some benefits for the RL problem would improve the contribution.
- To my knowledge, it appears that Algorithm 1 assumes knowledge of the true return distribution (or at least access to samples from it), which is a potentially unrealistic assumption. I may be misunderstanding however, so I listed this as a question below.

**Questions:**

- Is there a way to connect this method to the underlying RL problem?
- In Algorithm 1, how is $\mu_j(\mathcal{H}_N )$ evaluated? To my knowledge it seems we require knowledge of the true return distribution $\eta^\pi$ (or at the very least samples from it).

**Limitations:**

The authors discuss limitations, such as their method requiring access to IID samples.

---

> ### Author Rebuttal · Authors · 2023-08-09
>
> We thank the reviewer for the important points outlined.
>
> *The paper takes an RL agnostic approach, and learns an approximate return distribution without taking into account the underlying RL decision problem (i.e. Bellman operators aren't used, nor reward nor transition information, etc). This seems contrary to the decision-aware RL direction, which broadly states that we should focus learning capacity on aspects which are important for the decision problem. Some guarantees that learning a model in this way comes with some benefits for the RL problem would improve the contribution.
> Is there a way to connect this method to the underlying RL problem?*
>
> The proposed method is agnostic with respect to the nature of the underlying decision problem, which has advantages and disadvantages. Indeed, as the reviewer noted, we do not make use of Bellman operators, but just of return distributions, as commonly done in Monte Carlo approaches to RL. **While we agree that this might overlook some of the MDP structure, it has the advantage of leading to (i) a simpler mathematical treatment and (ii) of being applicable even to non-Markovian environments**. As the reviewer noted, we see this work as the first contribution taking "a step towards studying the interaction of distributional RL and representation learning". **Future works should for sure investigate the interaction between Temporal Difference (TD) distributional methods and representation learning**.
>
>
> *To my knowledge, it appears that Algorithm 1 assumes knowledge of the true return distribution (or at least access to samples from it), which is a potentially unrealistic assumption. I may be misunderstanding however, so I listed this as a question below.
> In Algorithm 1, how is $\mu_j(H_n)$ evaluated? To my knowledge it seems we require knowledge of the true return distribution (or at the very least samples from it).*
>
> The proposed algorithm does not require to evaluate $\mu_j(H_n)$, but rather $\hat \mu_j(\mathcal H_n)$, that corresponds to the empirical estimate of the value of the feature function $f_j$ as for Equation 6, computed from $N$ trajectory samples. In this sense,** it indeed requires trajectories of experience coming from the policy which is being evaluated but not the full knowledge of the environment transition models (and so of the return distribution)**.

---

> > ### Comment · Reviewer_W4d8 · 2023-08-10
> > **Response to rebuttal**
> >
> > I thank the authors for their response, and for clearing up my confusion regarding $\mu_j(\mathcal{H}_n)$. I am inclined to maintain my score.

---

### Official Review · Reviewer_Tf1V · 2023-07-27

**Soundness:** 3 good
**Presentation:** 3 good
**Contribution:** 3 good
**Rating:** 5
**Confidence:** 4

**Summary:**

The paper introduces a new framework called Distributional Maximum Entropy Policy Evaluation (D-Max-Ent PE) for policy evaluation in a distributional reinforcement learning setting. The framework considers the complexity of the representation used to evaluate a policy and incorporates it into a generalization-error bound. The authors then propose an algorithm called D-Max-Ent Progressive Factorization, which uses state-aggregation functions to progressively refine the state space representation while balancing information preservation and reduction in complexity. Numerical simulations demonstrate the effectiveness of the proposed algorithm and its relationship with aggregations and sample regimes.

**Strengths:**

This paper derived a practical formulation of the generalization error bound and proposed an algorithm, Distributional Max-Ent Progressive Factorization, that adaptively finds a feature representation to optimize the generalization error bound. Through illustrative simulations, the authors demonstrated the empirical behaviors of these approaches and explored the relationships between hyperparameters and the sampling regime.

**Weaknesses:**

1. This paper lacks experiments in practical reinforcement learning tasks, such as Atari. The authors should conduct more experiments to demonstrate the efficiency of the proposed policy evaluation method.

**Questions:**

None

---

> ### Author Rebuttal · Authors · 2023-08-09
>
> *This paper lacks experiments in practical reinforcement learning tasks, such as Atari. The authors should conduct more experiments to demonstrate the efficiency of the proposed policy evaluation method.*
>
> We thank the reviewer for this feedback. We remark that the main contribution of our work is of theoretical nature. Indeed, **the objective of the simulations was to illustrate two essential features of the proposed method that were only suggested by the theoretical result**, so as to close all the topics directly or indirectly introduced in the theoretical study:
>
> - whether it is possible to boost factorization by tuning the value of $\beta$,
> - whether the resulting factorization aggregated similar states in terms of the policy's true return distribution;
>
> rather than demonstrate its applicability to general reinforcement learning tasks. The MDP instances we employed were designed to be coherent with the objectives of the simulations and to be as much interpretable as possible in terms of true return distributions, a feature which is hardly shared with more complex RL tasks (e.g., Atari). We believe that the adaptation of the proposed method to more practical contexts is out of the scope of the present paper.

---

### Decision · Program_Chairs · 2023-09-21

**Decision:**

Accept (poster)

**Comment:**

This paper proposes a novel approach to distributional reinforcement learning (specifically policy evaluation) based on the maximum entropy principle.

The initial reviews for this paper were somewhat mixed. Reviewers generally agreed that the paper is clear, well written, and well organised. There was also general agreement that the core direction pursued by the authors (distributional policy evaluation via maximum entropy principle) is a novel, interesting direction, and that they provide rigorous theoretical analysis of the main algorithm introduced. Several reviewers also commented that this work represents an interesting route into studying the combination of distributional RL and representation learning, an area which is likely to be of interest to both RL and representation learning communities at NeurIPS.

There were also a few concerns expressed across reviews:
 - A lack of thorough experimentation in the paper. While the paper's primary contribution is purely theoretical, it is true that a wider range of experiments would strengthen the contribution. A related concern was that the experiments in the paper focus solely on tabular settings, without investigation of e.g. how applicable the core algorithm is in a setting such as the Atari suite.
 - A relatively RL-agnostic approach. The reviewers contribute a Monte Carlo learning algorithm, which does not make use of bootstrapping, and hence the theoretical analysis does not make use of the standard tools used in previous theoretical distributional RL work, such as Bellman operators.
 - The notion of representation learning used in the paper. The primary focus in the paper is on state aggregation, which is somewhat distinct from the notion of representation learning in deep learning; which is a form of representation learning commonly associated with distributional RL, since improved representation learning is a common hypothesis as to why distributional RL delivers benefits in combination with deep RL.

Different reviewers weighted these concerns differently in their reviews. After discussion, all reviewers except Reviewer wzKJ recommend acceptance for the paper, with the rebuttals also helping to resolve several minor questions raised by reviewers. Reviewer wzKJ has put forward their case clearly: that they weigh the concerns mentioned above highly, and although there has been good discussion between the authors and reviewer on these point, Reviewer wzKJ maintains their rejection recommendation. There has also been substantial discussion among reviewers on these points during the discussion period, as there is a wide range of opinion on the paper.

My overall recommendation for the paper, in light of discussion and review process described above, is for acceptance to NeurIPS. I understand Reviewer wzKJ's reservations, though I am comfortable that though not focused on temporal-difference learning, the authors' contribution represents an interesting approach to policy evaluation in distributional reinforcement learning. I would encourage the authors to incorporate the outcomes of discussion with all reviewers when preparing the camera-ready version of the paper.